# Selective N-terminal acylation of peptides and proteins with a Gly-His tag sequence

Manuel C. Martos-Maldonado[1,2], Christian T. Hjuler [1,2], Kasper K. Sørensen[1,2], Mikkel B. Thygesen [1,2], Jakob E. Rasmussen[1,2], Klaus Villadsen[1,2], Søren R. Midtgaard [3], Stefan Kol[4], Sanne Schoffelen[1,5] & Knud J. Jensen [1,2]

Methods for site-selective chemistry on proteins are in high demand for the synthesis of chemically modified biopharmaceuticals, as well as for applications in chemical biology, biosensors and more. Inadvertent N-terminal gluconoylation has been reported during expression of proteins with an N-terminal His tag. Here we report the development of this side-reaction into a general method for highly selective N-terminal acylation of proteins to introduce functional groups. We identify an optimized N-terminal sequence, GHHH$_n$— for the reaction with gluconolactone and 4-methoxyphenyl esters as acylating agents, facilitating the introduction of functionalities in a highly selective and efficient manner. Azides, biotin or a fluorophore are introduced at the N-termini of four unrelated proteins by effective and selective acylation with the 4-methoxyphenyl esters. This Gly-His$_n$ tag adds the unique capability for highly selective N-terminal chemical acylation of expressed proteins. We anticipate that it can find wide application in chemical biology and for biopharmaceuticals.

[1] Department of Chemistry, University of Copenhagen, Thorvaldsensvej 40, 1871 Frederiksberg, Denmark. [2] Biomolecular Nanoscale Engineering Center, University of Copenhagen, Thorvaldsensvej 40, 1871 Frederiksberg, Denmark. [3] Niels Bohr Institute, University of Copenhagen, Universitetsparken 5, 2100 Copenhagen, Denmark. [4] The Novo Nordisk Foundation Center for Biosustainability, Technical University of Denmark, Kemitorvet Building 220, 2800 Kgs. Lyngby, Denmark. [5] Center for Evolutionary Chemical Biology, University of Copenhagen, Universitetsparken 5, 2100 Copenhagen, Denmark. These authors contributed equally: Manuel C. Martos-Maldonado, Christian T. Hjuler, Kasper K. Sørensen, Mikkel B. Thygesen. Correspondence and requests for materials should be addressed to S.S. (email: sanne@chem.ku.dk) or to K.J.J. (email: kjj@chem.ku.dk)

Methods for site-selective modification of peptides and proteins are required for different fields, such as the development of biopharmaceutical conjugates (e.g., PEGylation, lipidation, and antibody–drug conjugates)[1], bioimaging[2], medical diagnostics[3], and material sciences[4]. Most proteins display multiple copies of the same side-chain at different locations. Reactions that target a particular functional group, e.g., the primary amine in Lys or thiol in Cys, potentially modify all occurrences of this residue, leading to the formation of a heterogeneous mixture of modified protein. Reactions involving proteins must also proceed in aqueous solutions under mild pH and temperature. These factors make it challenging to modify proteins in a regioselective manner. Several strategies have been developed to address this challenge. Some methods rely on genetic code expansion[5–7], which allows for the site-specific, ribosomal incorporation of non-canonical amino acids that exhibit suitable functionalities for bioorthogonal chemistry[8,9]. However, selective chemical methods applicable to proteins consisting of only canonical amino acids are an attractive alternative, as these proteins can be produced using standard, higher-yielding expression methods.

N-termini of proteins tend to reside on the surface of proteins[10] and are thus often well accessible to chemical modification[11]. Also, they can often be extended with additional amino acids without interference with protein function[11]. The N-terminal α-amine can be targeted for selective, pH-controlled acylation or alkylation due to its $pK_a$ value that is lower ($pK_a$ 7.6–8.0) than for Lys side-chain ε-amines ($pK_a$ 10.5 ± 1.1)[12]. Nevertheless, the selectivity is often challenged by the presence of a high number of competing ε-amines on Lys residues and the fact that their $pK_a$ can be lower due to the local environment. Also, direct acylation with NHS esters and reductive alkylation require a low-to-neutral pH. Alternatively, particular amino acids can be specifically targeted when located at the N-terminus, such as Cys via native chemical ligation[13] or 2-cyanobenzothiazole (CBT) condensation chemistry[14], Trp via Pictet–Spengler reactions[15,16], Ser and Thr via periodate oxidation to yield aldehydes for subsequent conjugation with α-nucleophiles[17–20], or Pro via oxidative coupling with aminophenols[11,21]. A small number of more general approaches for site-selective N-terminal modification without specific residue requirements have been described. A site-selective diazotransfer reaction for azide introduction has been achieved with imidazole-1-sulfonyl azide at pH 8–8.5[22], a phenyl ketene derivative has been used for alkyne introduction[23], a transamination reaction with 2-pyridinecarboxaldehydes has been reported[24], as well as reductive alkylation[25]. In addition to these methods for N-terminal labeling, the use of a four amino acid sequence (Phe-Cys-Pro-Phe) that enhances the reactivity of its cysteine residue for site-selective reaction with perfluoroaryl compounds has been reported[26]. Other small peptide sequences with high affinity for molecules of interest, e.g., fluorophores, have been reported. For instance, a tetracysteine tag (Cys-Cys-Xxx-Xxx-Cys-Cys) for selective reaction with an arsenic-modified fluorescein derivative[26,27] and a tetraserine motif (Ser-Ser-Pro-Gly-Ser-Ser) that binds a rhodamine-derived bisboronic acid[28].

Polyhistidine tags (His tags) are widely used for protein purification by immobilized-metal ion affinity chromatography[29]. However, it has been reported that during their expression His-tagged proteins can undergo N-terminal acylation with D-gluconic acid δ-lactone (GDL, 1) as an inadvertent side-reaction[30]. Geoghegan et al. observed gluconoylation of proteins when expressed in *Escherichia coli*, while using His tags with the N-terminal sequence GSSHHHHHHSSGLVPR–. They also reported that synthetic peptides GSSHHHHHHSSGLVPR, GSAHHHHAAR, GASHHHHAAR, and GAAHHHHAAR could be modified with GDL in HEPES buffer at pH 7.5[30].

Here, we convert an undesirable side-reaction into a highly selective chemical method for modification of peptides and proteins (Fig. 1). We identify an optimal N-terminal sequence (GHHH$_n$) using GDL as inexpensive, water-soluble acylating agent. Studying the reactivity with other acylating agents, 4-methoxyphenyl esters gave good selectivity for acylation of this N-terminal tag. 4-Methoxyphenyl esters facilitate the introduction of small reactive groups, such as azides, or the direct conjugation of functional molecules such as biotin. We demonstrate the potential on several peptides and proteins. The GHHHHHH tag, which can be fused to the N-terminus of any protein of interest, offers a dual functionality His tag as it can still be used for affinity purification. We believe that this methodology will be a valuable contribution to the toolbox for N-terminal modification. We propose the name His tag acylation to describe this method.

## Results

**D-Gluconic acid δ-lactone for peptide modifications**. We hypothesized that highly site-selective gluconoylation of the Nα-amine could be achieved with an optimized N-terminal sequence. We synthesized 11 peptides with different N-terminal tags attached to the sequence LRFKFY-NH$_2$ (Table 1). All peptides included one Lys residue to test the selectivity of the reaction for the N-terminal α-amine. The peptides were treated at a concentration of 1 mM in 200 mM HEPES buffer at pH 7.5 with 25 equiv. of GDL (1) at room temperature.

The base peptide sequence LRFKY-NH$_2$ (2) by itself, which has an N-terminal Leu and a Lys with an ε-amino side-chain, was practically resistant to N-terminal acylation (Table 1). Addition of a Gly residue with a non-hindered α-amine (3, 4) increased the degree of N-acylation somewhat. Addition of N-terminal His residues to give peptides 5 and 6 significantly improved N-acylation. Also, the presence of three His was beneficial (5 vs 6). N-terminal addition of Gly to His improved the N-acylation (5 vs 7) further, while a Ser[30] in between Gly and His was detrimental (7 vs 8). Increasing the number of His from Gly-His to Gly-His-His-His (4, 9 vs 10) provided a clear improvement in propensity to be acylated by GDL. Additional control experiments confirmed the importance of a sterically non-hindered Gly α-amine N-terminally of the His-His-His, as corresponding peptides with N-terminal Ala and Val proved increasingly resistant to N-acylation (6, 11, 12 vs 10).

Formation of di-gluconoylated products of these peptides was not observed in any case, which pointed towards a site-selective modification of the N-terminal α-amine. To further substantiate this, the selectivity of the reaction was analyzed in the case of peptide 9. NMR spectroscopy confirmed that the reaction took place selectively at the N-terminal α-amine and not on the ε-amine of the Lys residue (Supplementary Fig. 1).

Encouraged by these observations, we evaluated the optimized GHHH sequence in the 18-mer peptide 13 (DWLKAFYDK-VAEKLKEAF, Beltide-1), which has four Lys residues. Beltide-1 is an amphipathic helical peptide that self-assembles to give lipid filled nanodiscs[31]. The obtained results with model peptides 2–12 suggested that the optimal N-terminal sequence for site-selective gluconoylation was GHHH$_n$– (where $n \geq 1$). Commonly used His tags for protein purification by immobilized-metal ion affinity chromatography can be placed N- or C-terminally and typically contain six His residues[29]. We hypothesized that adding extra His residues to GHHH could improve the ability to be acylated. We compared GHHH- with GHHHHHH-tags on a Beltide-1 peptide sequence (Table 2). First, control experiments with non-tagged Beltide-1 (13) and its acetylated analog (13Ac) did not show any peptide acylation. Next, GHHH-Beltide-1 (14) was N-acylated in

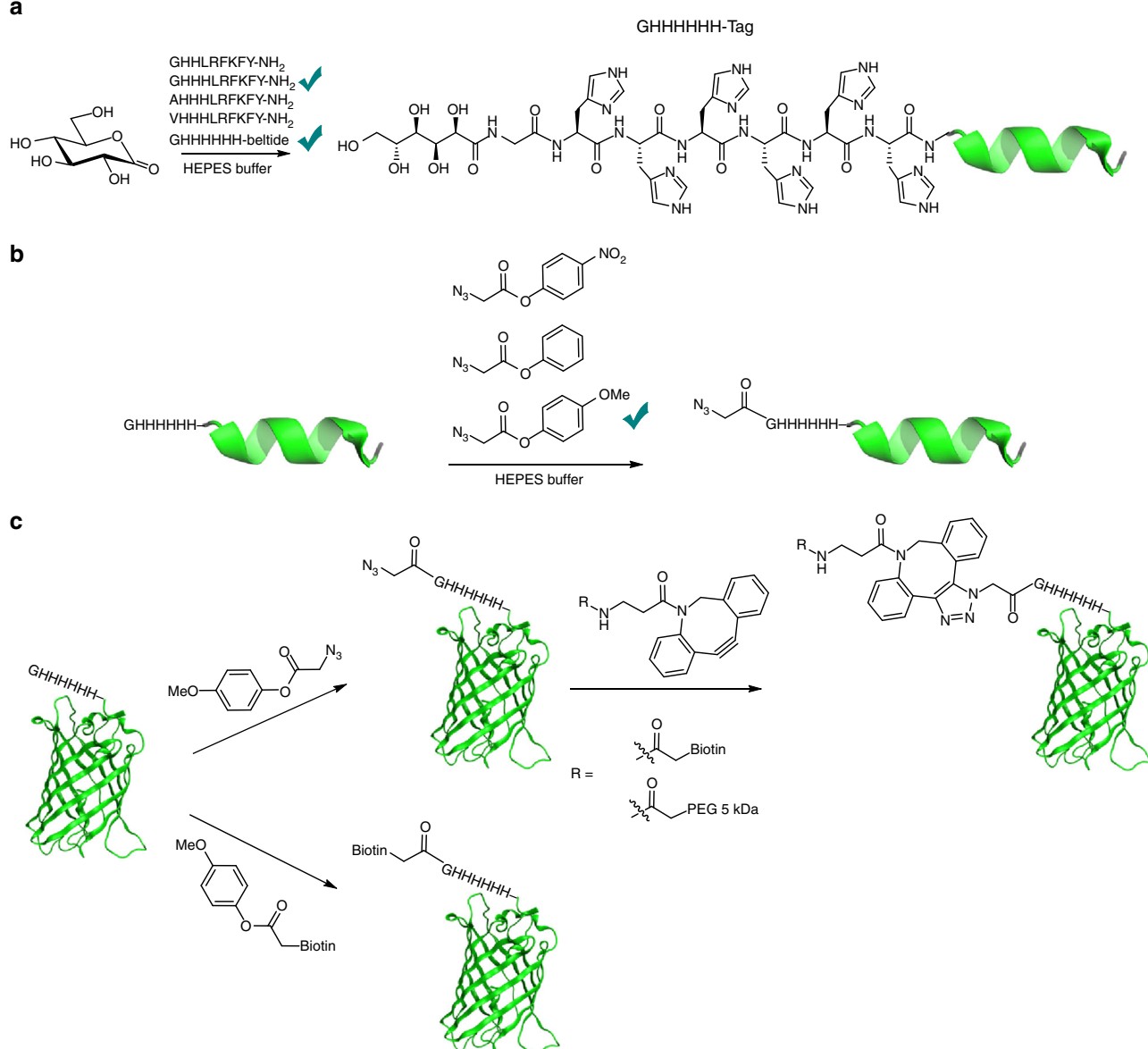

**Fig. 1** Concept for N-terminal His tag acylation of Gly-His$_n$ sequences. **a** Optimization of the His tag sequence for acylation with GDL; **b** identification of 4-methoxyphenyl esters as functional acylating agents; and **c** application of the His tag acylation to proteins

73%, while GHHHHHH-Beltide-1 (**15**) showed higher reactivity and reached full conversion to the corresponding mono-gluconoylated derivative after 1 h, when treated at a concentration of 1 mM with 100 equiv. of GDL in 200 mM HEPES buffer at pH 7.5 at room temperature (Supplementary Fig. 2). Remarkably, formation of di-gluconoylated peptides was not detected. Importantly, in a control experiment we observed that N-terminally acetylated peptide Ac-GHHHHHH-Beltide-1 (**15**) was not acylated by GDL, which again indicated that the reaction occurred selectively at the N-terminal α-amine, when available.

**Phenyl esters as acylating agents.** While other acylating agents, including lactones, thioesters, and a N-hydroxysuccinimide ester were also tested (Supplementary Fig. 3), they only gave limited N-terminal acylation or low selectivity. We then focused on tuning the properties of representative phenyl esters **16–18** (Fig. 2). The 4-nitrophenyl and unsubstituted phenyl esters proved too reactive and were not selective in the acylation, whereas 4-methoxyphenyl

ester **18** provided good selectivity while retaining reactivity. The half-life of **18** in 200 mM HEPES buffer, pH 7.5, at 4 °C was 3.8 h (Supplementary Fig. 4).

When GHHHHHHH-Beltide-1 (**15**) at 1 mM was reacted with 2.5 equiv. of **18** at 4 °C for 24 h, full conversion was achieved (92% mono-acylated (**15Az**) and 8% di-acylated peptide) (Table 3, Supplementary Fig. 5 and 6). Shorter reaction time or lower equiv. led to exclusive formation of mono-acylated peptide, although with lower conversion. The same conditions were tested on other Beltide-1 derivatives (Table 3). GHHHHHH-Beltide-1 (**15**) and GHHHHHHG-Beltide-1 (**15G**) showed higher reactivity and selectivity than GHHH-Beltide-1 (**14**). The beltide sequence has four Lys (position 4, 9, 13, and 15). Trypsin digestion of the diacylated product (**15Az-Az**) showed that the second acylation had occurred chiefly at Lys-4, located most closely to the His acylation tag (Supplementary Fig. 7 and 8). The small difference between **15** and **15G** could point to a minimal effect of the additional spacing of Lys-4 and the His acylation tag provided by the Gly, in this particular case.

**Table 1 Acylation of a series of peptides with GDL**

| Peptide | Peptide sequence | Acylation (%)[a] |
|---|---|---|
| **2** | LRFKFY-NH$_2$ | 3.0 ( ± 0.7) |
| **3** | **G**LRFKFY-NH$_2$ | 17.3 ( ± 0.7) |
| **4** | **GS**LRFKFY-NH$_2$ | 15.7 ( ±1.6) |
| **5** | **H**LRFKFY-NH$_2$ | 29.8 ( ± 0.7) |
| **6** | **HHH**LRFKFY-NH$_2$ | 64.4 ( ± 13.9) |
| **7** | **GH**LRFKFY-NH$_2$ | 45.2 ( ±1.7) |
| **8** | **GSH**LRFKFY-NH$_2$ | 33.4 ( ±1.9) |
| **9** | **GHH**LRFKFY-NH$_2$ | 63.7 ( ± 8.6) |
| **10** | **GHHH**LRFKFY-NH$_2$ | 81.9 ( ± 3.5) |
| **11** | **AHHH**LRFKFY-NH$_2$ | 27.4 ( ± 4.3) |
| **12** | **VHHH**LRFKFY-NH$_2$ | 4.3 ( ± 7.5) |

[a]Conversion of peptides **2–12** (1 mM) after treatment with 25 equiv. of GDL (**1**) in 200 mM HEPES buffer at pH 7.5 at room temperature for 2 h, as determined by LC–MS (UV, 215 nm). Standard deviations are based on triplicate measurements. The His acylation tags and compound numbers are indicated in bold

**Table 2 Acylation of a series of Beltide-1 derivatives with GDL**

| Peptide | Acylation (%)[a] |
|---|---|
| Beltide-1 (**13**) | No reaction |
| **Ac**-Beltide-1 (**13Ac**) | No reaction |
| **GHHH**-Beltide-1 (**14**) | ~ 73 |
| **GHHHHHH**-Beltide-1 (**15**) | 100 |
| **Ac-GHHHHHH**-Beltide-1 (**15Ac**) | No reaction |

[a]Conversion of Beltide derivatives (1 mM) after treatment with 100 equiv. of GDL (**1**) in 200 mM HEPES buffer at pH 7.5 at room temperature for 1 h, as determined by LC–MS (UV, 215 nm). The His acylation tags and compound numbers are indicated in bold

In **15K** and **15K-Ac** an additional Lys was placed C-terminal of the GHHHHHH sequence to study the selectivity with a Lys Nε-amine close to His acylation tag. Treatment of **15K** with **18** gave a slightly elevated level of diacylation (14%), while for N-terminally blocked 15K-Ac an acylation of 31% was observed. This showed that while the His acylation tag can promote reaction at a neighboring Lys Nε-amine it has a high selectivity for N-terminal acylation.

**Protein modifications**. The optimized N-terminal sequence GHHHHHH (GH$_6$) and the conditions for acylation were then evaluated on proteins. First, we employed GDL to modify three GH$_6$-tagged proteins, enhanced green fluorescent protein (EGFP, 30.8 kDa), maltose binding protein (MBP, 41.0 kDa) and a small ubiquitin-related modifier protein (SUMO, 12.5 kDa). EGFP is known for its intrinsic fluorescence and is widely used as model protein. MBP, a protein which is part of the maltose/maltodextrin system of *E. coli*, is commonly used to increase the solubility of recombinant proteins by generation of an MBP-protein fusion. SUMO proteins, structurally related to ubiquitin, modify the function of the proteins to which they are conjugated. Like MBP, they are also used as fusion tag to enhance expression and solubility of recombinant proteins. The three model proteins contain 21, 35, and 9 Lys residues, respectively. The GH$_6$-tagged proteins were successfully expressed in *E. coli*. Both EGFP and MBP were purified by Ni$^{2+}$ affinity chromatography, which confirms that this GH$_6$-tag still functions as purification tag. Rewardingly, the reactions with GDL yielded only the mono-functionalized proteins in all cases, as can be seen in Fig. 3a.

As determined by MALDI-TOF MS, trypsination of gluconoy-lated GH$_6$-EGFP yielded the gluconoylated N-terminal fragment (amino acids 1–14) (Fig. 3b). Only the unmodified version of tryptic fragment #29–276 was detected by ESI-TOF MS, indicating that none of the 20 Lys residues located in this fragment were modified. Furthermore, digestion with chymo-trypsin showed that fragment 20–29 ($m/z = 1158.623$) was detected but not its gluconoylated version, which revealed that Lys21, the only Lys residue in the first 28 amino acids of the protein, had remained unmodified. Taken together, these observations indicated that the modification with GDL occurred exclusively at the N-terminus of EGFP.

Interestingly, for both **15** and GH$_6$-EGFP the gluconoylation proved to be reversible (Supplementary Fig. 9 and 10). We hypothesize that the reversibility can be due to the polyol structure in the gluconoylated protein, however, this would require additional studies. In contrast, the acylation with 4-methoxyphenyl esters of acetic acid derivatives gave stable products (Supplementary Fig. 11 and 12).

For comparison, additional experiments were carried out with GSSH$_6$-EGFP, which has a conventional His tag, as well as non-tagged GSH-EGFP. In both cases, some formation of mono-gluconoylated product was observed, although with significantly lower conversion in comparison to GH$_6$-EGFP (Fig. 3c). These results highlighted the importance of our His acylation tag for the reaction to occur. GH$_6$-EGFP and GSSH$_6$-EGFP are identical except for the two inserted Ser residues; thus, these data confirmed the importance of an uninterrupted N-terminal Gly-His sequence.

Next, the 4-methoxyphenyl ester **18** was used to modify GH$_6$-EGFP. Gratifyingly, **18** was effective in labeling the protein at 4 °C. When a 35 μM solution of GH$_6$-EGFP in 200 mM HEPES buffer at pH 7.5 was treated with 40 equiv. of **18** for 4 days, followed by the addition of two aliquots of 10 equiv. of **18** in the next two days, 88% conversion was observed by ESI-MS (71% conversion to mono-functionalized products) (Fig. 4b). Here it should be noted that the 4 days incubation time can be shortened to one day. In follow-up experiments and in agreement with the observed half-life of **18** at 4 °C (see above), it was observed that no additional acylation occurred after one day, unless a fresh portion of phenyl ester **18** was added. In a second scenario for optimization, fewer equivalents and shorter reaction times resulted in a somewhat lower overall conversion, but also in a higher selectivity towards the mono-functionalized species. For example, when the protein was treated with 20 equiv. of **18** for 1 day, 51% was converted and 45% was mono-functionalized (Supplementary Fig. 13).

Azido-functionalized N$_3$-CH$_2$-C(O)-GH$_6$-EGFP was reacted with DBCO-OEG$_4$-biotin (**19**, Fig. 4) to demonstrate the potential of the strategy for subsequent strain-promoted alkyne–azide conjugation. ESI-MS analysis displayed predominant formation of the mono-labeled protein (Fig. 4c). The sample was digested with trypsin and subsequently analyzed by MALDI-TOF MS. The labeled N-terminal fragment (amino acids #1–14) was observed (Supplementary Fig. 14). Furthermore, as observed for gluconoy-lated GH$_6$-EGFP (Fig. 3b), a peak corresponding to unmodified fragment #29–276 was present, while the labeled version was not detected. This indicated that no modification took place at this region of the protein. Azido-functionalized N$_3$-CH$_2$-C(O)-GH$_6$-EGFP was also conjugated to a 5 kDa DBCO-PEG. SDS-PAGE analysis revealed formation of mono-PEGylated protein with high conversion, while di-PEGylated species were only observed as very faint bands (Fig. 4d).

For comparison, the potential reaction between compound **18** and GSH-EGFP and GSSH$_6$-EGFP were studied. The former displayed negligible formation of the mono-acylated species when treated at 35 μM with 40 equiv. of **18** in 200 mM HEPES buffer at pH 7.5 with 10% acetonitrile for 2 days. Under the same

**Fig. 2** Evaluating substituted phenyl esters for His tag acylation. Phenyl esters **16–18**

**Table 3 Acylation of a series of Beltide-1 derivatives with 4-methoxy phenyl ester 18[a]**

| Peptide | N-Terminal mono-acylated product (%) | Di-acylated product (%) |
|---|---|---|
| Beltide-1 (**13**) | No reaction | No reaction |
| **Ac**-Beltide-1 (**13-Ac**) | — | No reaction |
| **GHHH**-Beltide-1 (**14**) | 63 | 13 |
| **GHHHHHH**-Beltide-1 (**15**) | 92 | 8 (±1) |
| **Ac-GHHHHHH**-Beltide-1 (**15Ac**) | — | 9 (±0.5) |
| **GHHHHHHG**-Beltide-1 (**15G**) | 97 | 3 (±0.5) |
| **GHHHHHHK**-Beltide-1 (**15K**) | 87 | 13 (±1) |
| **Ac-GHHHHHHK**-Beltide-1 (**15K-Ac**) | — | 31 (±4) |

[a]Conversion of a series of Beltide-1 derivatives after being treated at a concentration of 1 mM with 2.5 equiv. of 4-methoxy-phenyl ester **18** at 4 °C for 24 h as determined by LC-MS (UV, 215 nm). Standard deviations are based on duplicate measurements. Beltide-1 sequence DWLKAFYDKVAEKLKEAF, Lys underscored. The His acylation tags and compound numbers are indicated in bold

conditions, GSSH$_6$-EGFP was modified, although with lower selectivity than GH$_6$-GFP. Mono-acylated protein was still observed as the major species in ESI-MS, but also significant amounts of di-, and even tri-acylated species (Supplementary Fig. 15). The low selectivity can be explained by the increase in the distance between the N-terminal α-amine and the imidazole side-chains of the His tag. This again confirmed GHHHHHH as an optimal N-terminal sequence for acylation. Importantly, it was observed that neither Ac-GHHHHHH-NH$_2$ nor imidazole catalyzed the N-terminal acylation of GSH-EGFP by **18** (Supplementary Fig. 16).

To further evaluate the generality of the selective acylation of the N-terminal segment, we reacted GH$_6$-tagged MBP with 4-methoxyphenyl ester **18**. Rewardingly, under the conditions tested mono-functionalized products were the major species (35 μM protein, 36 equiv. **18**, 4 °C). Conversion to mono-labeled protein was determined by ESI-MS as 59% for GH$_6$-MBP (Fig. 4e).

To exemplify the optimization of the selective N-terminal acylation, we reacted GH$_6$-tagged SUMO with 4-methoxyphenyl ester **18**. Under all conditions tested mono-functionalized products were the major species (Supplementary Table 1, Fig. 4e). Parameters (Supplementary Table 1) such as numbers of equivalents, temperature, and time can be used to optimize the acylation of a protein, in this case GH$_6$-SUMO. The multiple additions of ester **18** took its half-life in aq. solution into consideration. Conversion to mono-labeled GH$_6$-SUMO was determined by ESI-MS to be up to 80% (155 μM protein, 10 + 10 equiv. **18**, 3 + 6 h, 25 °C).

In order to demonstrate the application of the method on a biologically more relevant protein, we selected X-linked inhibitor of apoptosis protein (XIAP) as fourth model protein. The inhibitor of apoptosis proteins (IAPs) constitute a class of proteins that play an essential role in the anti-apoptotic and pro-survival signaling pathways. They are characterized by having one or more baculovirus IAP repeats called BIR domains. These domains of ~70 amino acids interact with different kinds of proteins. IAPs are upregulated in various cancers and associated with tumor growth and resistance to treatment. Therefore, IAPs are attractive targets for antitumor drug discovery[32]. The site-selective, N-terminal introduction of a fluorophore or biotin

moiety in XIAP will facilitate studies aiming at the identification and characterization of interaction partners or inhibitors of this class of proteins. Here, the XIAP fragment [124–240] was selected which comprises the linker region between BIR1 and BIR2 and the BIR2 domain, and which is known to potently inhibit caspase-3 and -7[33].

A GH$_6$-tagged version of the protein (for simplicity referred to as GH$_6$-BIR2) was expressed, purified by Ni$^{2+}$ affinity chromatography, and treated with compound **18**. It was found that treatment with 20 equiv. of compound **18** was optimal, leading to 83% conversion and 65% mono-functionalization (Supplementary Fig. 17). In order to test whether acylated GH$_6$-BIR2 had retained its function, a caspase-7 inhibition assay was performed using the tetrapeptide DEVD linked to 7-amino-4-trifluoromethylcoumarin (AFC) as substrate. Acylated GH$_6$-BIR2 was as potent in inhibiting caspase-7 as its non-modified counterpart (Fig. 5b). Next, the possibility to fluorescently label azido-functionalized N$_3$-CH$_2$-C(O)-GH$_6$-BIR2 through the Cu (I)-catalyzed alkyne–azide cycloaddition was demonstrated, as depicted in Fig. 5c. Furthermore, in order to evaluate whether 4-methoxyphenyl esters could be used to directly attach larger moieties, biotin derivative **20** (Fig. 5d) was reacted with GH$_6$-BIR2. Treatment with 20 equiv. of compound **20** at room temperature for 24 h led to a ~50% conversion to mono-labeled protein as determined by ESI-MS. Addition of an extra aliquot of 20 equiv. for 1 more day gave 80% conversion with 60% mono-functionalization. The ability of biotinylated GH$_6$-BIR2 to bind streptavidin was demonstrated by western blot (Fig. 5d).

As an alternative approach, we tested whether GH$_6$-EGFP could be biotinylated with a 4-methoxyphenyl ester derivative of biotin formed in situ by coupling biotin reagent **19** to compound **18** via the strain-promoted alkyne-azide cycloaddition. Approximately 45% mono-labeled protein was obtained when GH$_6$-EGFP was incubated with 60 equiv. of the in situ formed ester for 48 h at 4 °C (Supplementary Fig. 18).

GH$_6$-SUMO was found to react with compound **20** to a similar degree and with comparable selectivity as GH$_6$-BIR2. Moreover, biotinylated SUMO could still be processed by SUMO protease as confirmed by ESI-MS (Supplementary Fig. 19). This result suggests that also other enzymes involved in the SUMOylation process will still recognize the modified SUMO.

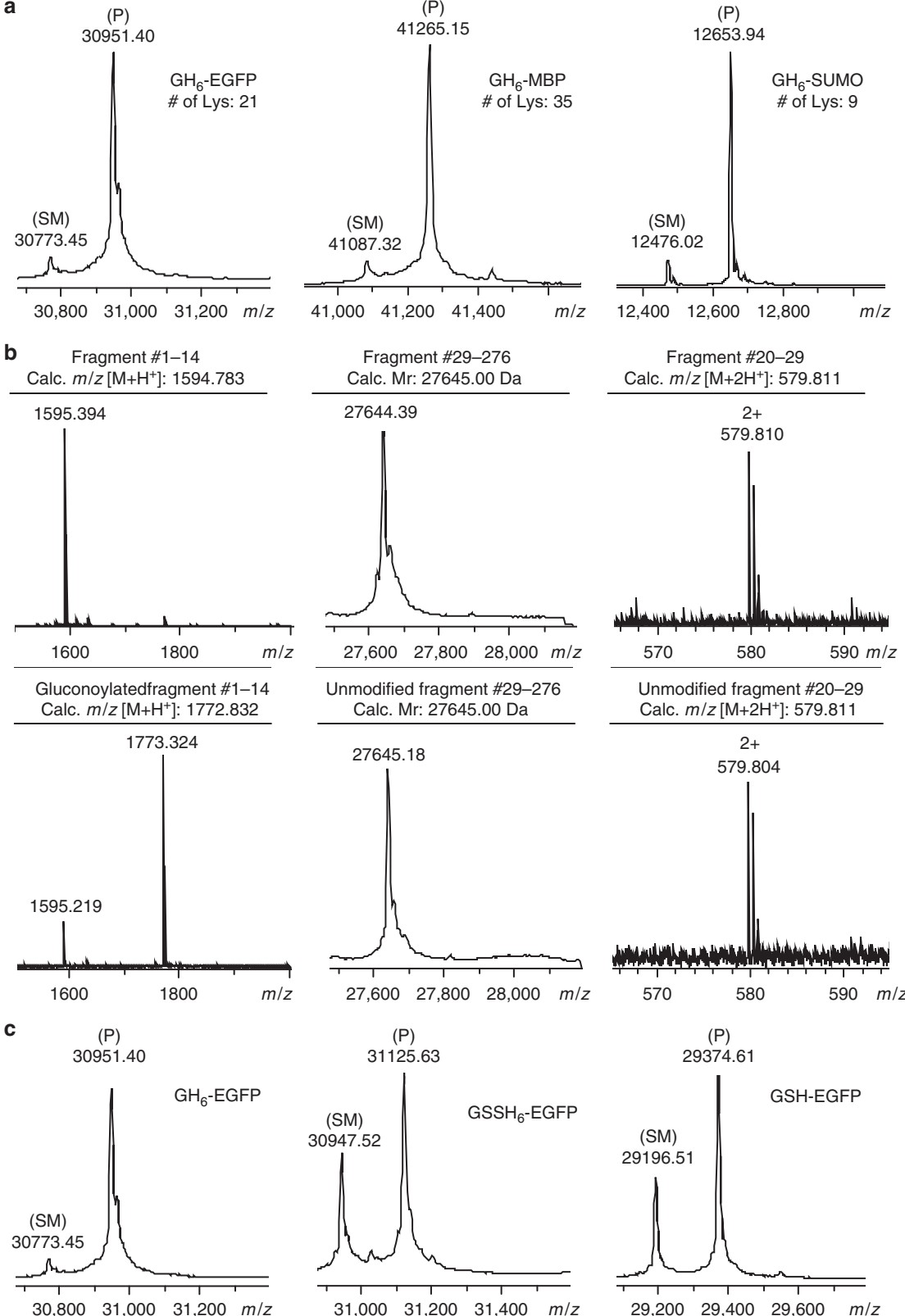

**Mechanistic studies.** The Gly-His$_n$ sequence catalyzes the acylation of the Gly Nα-amine. Fundamental studies of acyl transfer catalysis by imidazoles have shown that the catalytic mechanism is modulated by variations in the leaving group of the acylating agent[34–40]. Strong electrophiles, such as acetic anhydride and 4-nitrophenyl acetates, favor a nucleophilic mechanism of

imidazole catalysis involving intermediate formation of neutral N-acyl imidazoles. In contrast, weak electrophiles, such as alkyl esters, react by a specific base catalytic mechanism. Interestingly, 4-methoxyphenyl acetate is known to be a borderline case between nucleophilic and specific base catalysis[40]. Both of the above mentioned types of mechanisms may be envisioned in

**Fig. 3** Site-specific modification of proteins with GDL. **a** Deconvoluted ESI-TOF spectra of the reaction of GDL with GH$_6$-tagged EGFP, MBP, and SUMO. The GH$_6$-tagged proteins were reacted in 200 mM HEPES buffer at pH 7.5 and room temperature. GH$_6$-EGFP (350 μM) was treated with 200 mM lactone for 1 h. GH$_6$-MBP (350 μM) was treated with 350 mM lactone for 4 h. GH$_6$-SUMO (42 μM) was treated with 189 mM lactone for 1 h. **b** Detection of N-terminal selective gluconoylation of GH$_6$-EGFP. MS spectra of non-reacted and gluconoylated GH$_6$-EGFP after digestion with trypsin or chymotrypsin are depicted. MALDI-TOF spectra of the N-terminal fragment consisting of amino acids #1–14, deconvoluted ESI-TOF spectra of tryptic fragment #29–276, and ESI-TOF spectra of chymotryptic fragment #20–29 containing Lys21 (the only Lys residue in the first 28 amino acids of the protein) are depicted. The peak at 1595.219 Da detected in the MALDI-TOF spectrum of the gluconoylated protein was attributed to the partial reversibility of this reaction, which was also observed for peptide **15** (Supplementary Fig. 9 and 10). **c** Deconvoluted ESI-TOF spectra of the reaction of three EGFP variants with GDL. The proteins were reacted at a concentration of 35 μM with 200 mM GDL for 1 h in 200 mM HEPES buffer at pH 7.5 and room temperature. The degrees of conversion, based on the deconvoluted MS data, were 92%, 60%, and 66% for GH$_6$-EGFP, GSSH$_6$-EGFP, and GSH-EGFP, respectively. Unmodified proteins are labeled SM, and species corresponding to the correct product mass are labeled P

the present case of His tag acylation. Initial interaction with the imidazole ring or base catalysis seems to be a requirement, since no reaction was observed in control experiments with non-tagged Beltide-1 (**13**) or its acetylated derivative **13Ac**. In the case of a hypothesized nucleophilic catalysis mechanism, the reaction would start by formation of an N-acyl imidazole intermediate by reaction of the ester with the imidazole side-chains of the His residues. Then, the acyl imidazole could react with amines or undergo hydrolysis. When the N-terminal Gly α-amine is available the reaction would occur preferentially on this amine, which can be explained by its closer proximity and lower p$K_a$ in comparison with amines of Lys side-chains. Alternatively, for a specific base catalytic mechanism, the imidazoles of the His side-chains would assist in deprotonation of the charged, tetrahedral addition complex between the acylating agent and the N-terminal Gly α-amine. Once again, the close proximity of the His side-chain would further promote catalysis. In order to elucidate the mechanism, we conducted a set of control experiments where the reactivity of 4-methoxyphenyl ester **18** and acetic anhydride (as positive control) with different imidazoles was studied at pH 7.5. The hypothetical N-acyl derivatives of imidazole or Ac-GHHHHHH-NH$_2$ were not observed in any experiments involving **18**, neither by UV (245 nm)[40] or $^1$H-NMR spectroscopy. Interestingly, nor was the N-acyl imidazole derivative detectable in the presence of the more reactive 4-nitrophenyl or phenyl esters, **16** or **17**, under the applied reaction conditions. In contrast, full conversion to N-acetylimidazole was evident when acetic anhydride was used as the acylating agent (Supplementary Fig. 20 and 21).

Furthermore, we compared hydrolysis rates of **18** in the presence of imidazole or 2-isopropylimidazole in order to reveal the involvement of any transiently formed N-acyl imidazole. The steric hindrance of the 2-isopropyl group is known to affect nucleophilic catalytic processes negatively, whereas specific base catalytic processes of 2-isopropylimidazole are virtually unaffected, as compared to imidazole[34,35]. The observed hydrolysis rate of **18** with 6 mM 2-isopropylimidazole was essentially identical to that of 6 mM imidazole or 1 mM Ac-GHHHHHH-NH$_2$ in HEPES buffer, pH 7.5 (Supplementary Fig. 22).

## Discussion

We present a chemical method for highly selective and efficient N-terminal acylation of proteins. This method, for which we propose the name His tag acylation, is based on a short N-terminal peptide sequence, GHHHHHH, and is complemented by the use of 4-methoxy phenyl esters as finely tuned acylating agents. The method proceeds in aqueous medium, at mild temperature and neutral pH. In comparison with the oxidation of an N-terminal Ser or Thr residue or sortase-mediated N-terminal modification, the reaction avoids the use of harsh oxidants or expensive enzymes. Moreover, the method, which was successfully applied to four unrelated proteins, is anticipated to be highly

versatile. Structural variation among proteins is known to affect the yield of chemical N-terminal modification strategies, such as the transamination reaction using pyridoxal-5′-phosphate or 2-pyridinecarboxyaldehyde and the diazotransfer reaction. In contrast to these strategies, the hydrophilic nature of our GHHHHHH sequence will contribute to an increased accessibility of the N-terminus and we anticipate that it can provide a high yield over a wide range of proteins. Furthermore, oxidative methods require two steps for introducing an azide. The Gly-His$_n$ sequence will remain in the protein after acylation, however, the sequence can be rather short. Other methods for N-terminal modification, such as sortase mediated reactions, also rely on short peptide sequences, which remain in the final protein conjugate.

The optimized sequence constitutes a unique version of one of the most widely used affinity tags for protein purification, the His tag. In commonly used bacterial expression vectors, N-terminal His tags are typically preceded by 3–4 amino acids. Here, we show that by direct attachment of the HHHHHH segment to an N-terminal Gly residue instead, a tag with dual functionality is generated, facilitating both purification and efficient, site-specific modification of recombinant proteins. The non-hindered N-terminal α-amine of Gly guarantees high conversion, while the proximity of the His segment directs the selectivity of the reaction. As demonstrated, proteins containing the GH$_6$ tag can easily be obtained by bacterial expression. One can rely on the excision of the N-terminal methionine, which is known to occur during expression when the second amino acid is small. Alternatively, in case an N-terminal signal peptide is required, for example, for secretion to the periplasm, this signal peptide can be removed using the commonly employed TEV protease, thereby revealing the needed N-terminal Gly.

Our results indicate that the mechanism behind the very high selectivity of the His tag acylation was specific base catalysis, in which a His side-chain assists deprotonation during the direct acylation of the Gly α-amine. Facilitation of this type of proton transfer is a key driving force for reaction rate enhancements in aqueous, neutral medium[41], including for enzymatic reactions. The ester reacts preferentially with assistance from His side-chain imidazoles since they are not protonated (p$K_a \sim 6.0$) at the pH of the reaction, in contrast with the N-terminal α-amine (p$K_a \sim 7.6$–8.0) and Lys side-chains (p$K_a \sim 10.5$). The presence of the additional five His residues in the His tag may serve to modulate the basicity of the imidazole nitrogen of the catalytic residue. A recent study has shown that the p$K_a$ values of individual His side-chains in a His$_6$ tag span a range from 4.8 to 7.5[42]. This implies that imidazoles with different acid-base properties are to some extent organized within the His tag. A His side-chain in an α-helix can in some cases catalyze the acylation of a flanking Lys Nε-amine in position $i-3$ and $i+4$[43]. However, this requires an engineered α-helix, whereas the present method provides a general method for N-terminal Nα-amine acylation.

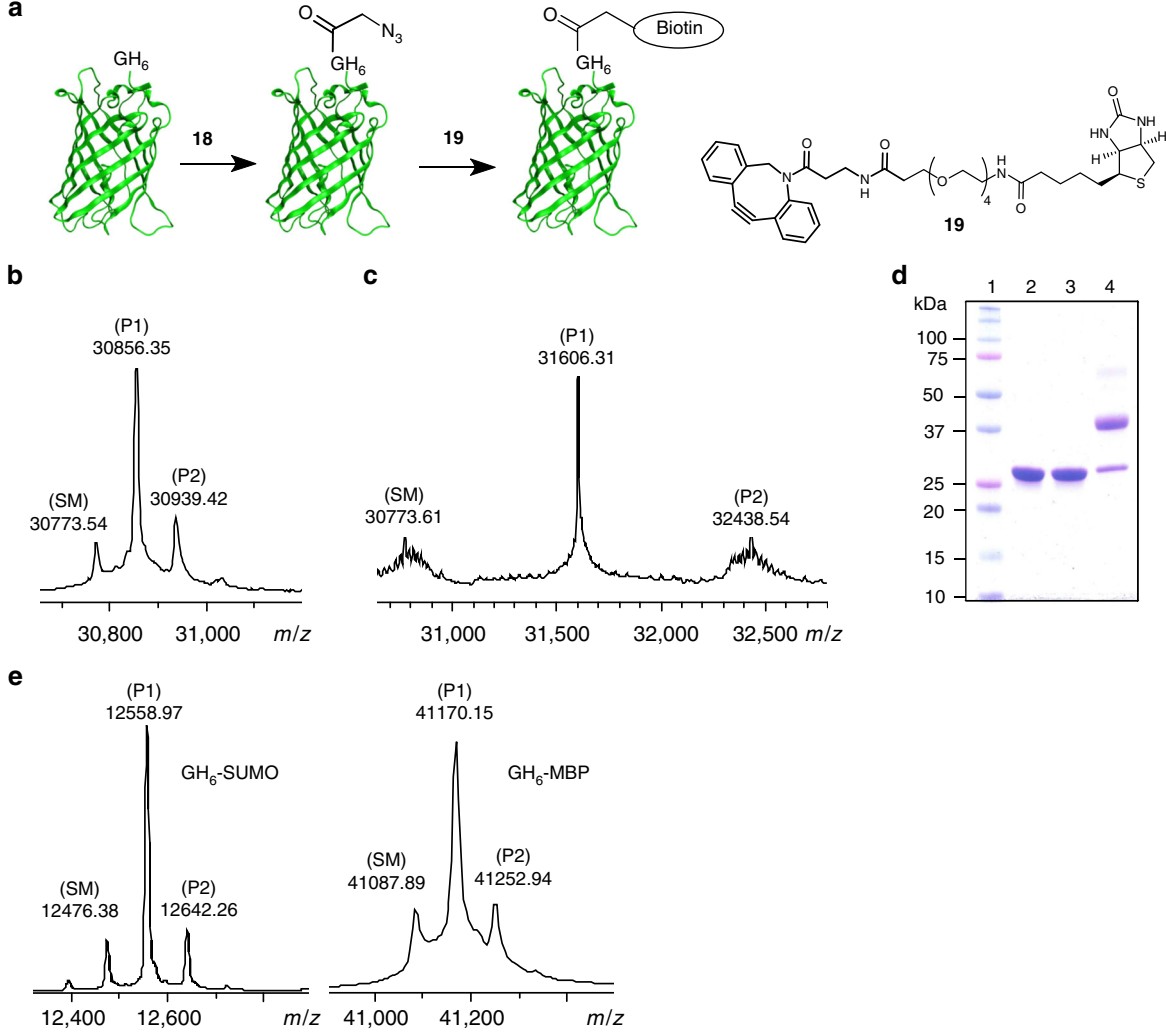

**Fig. 4** Functionalization of proteins with azides, biotin and PEG. **a** Schematic representation of the two-step biotinylation of GH$_6$-EGFP. Reaction conditions for the biotinylation step: 100 µM azido-functionalized N$_3$-CH$_2$-C(O)-GH$_6$-EGFP with 1.5 equiv. of **19** in 100 mM phosphate buffer with 1% acetonitrile at pH 7.5 and room temperature for 1 h. **b** Deconvoluted ESI-TOF spectrum of the reaction of GH$_6$-EGFP with phenyl ester **18**. **c** Deconvoluted ESI-TOF spectrum of the reaction of azido-functionalized GH$_6$-EGFP with DBCO-PEG4-biotin **19**. **d** SDS-PAGE analysis of the two-step PEGylation of GH$_6$-EGFP. Lane 1: protein marker, lane 2: GH$_6$-EGFP, lane 3: azido-functionalized GH$_6$-EGFP, lane 4: reaction of azido-functionalized GH$_6$-EGFP with 5 kDa DBCO-PEG. Reaction conditions for the PEGylation step: 100 µM azido-functionalized N$_3$-CH$_2$-C(O)-GH$_6$-EGFP with 2.5 equiv. of 5 kDa DBCO-PEG in 100 mM phosphate buffer at pH 7.5 and room temperature for 3 h. **e** Deconvoluted ESI-TOF spectra of the reactions of GH$_6$-SUMO (Supplementary Table 1, Entry 8) and GH$_6$-MBP with phenyl ester **18**. Unmodified proteins are labeled SM, and species corresponding to the product masses are labeled P1 and P2 (the number indicating the number of azides or biotins, respectively)

4-Methoxyphenyl esters, such as azido reagent **18** and biotin reagent **20**, were found to provide the best performance in terms of reactivity vs selectivity. These reagents are easy to prepare and allow for the one-step introduction of an N-terminal functionality. As demonstrated, azido acetyl moieties can be smoothly introduced and used for subsequent azide–alkyne couplings. Functional moieties, such as a biotin, can be incorporated in a single step as well. Importantly, the reactivity of the acylating agent and the catalytic properties of the His tag are optimized for selective reaction at the N-terminus. The selectivity for N-terminal acylation of proteins with our His acylation tag should be seen in view of the number of competing Lys residues. EGFP has 21 Lys residues, MBP has 35, SUMO 9 and Bir2 3. For example, a 9:1 ratio for acylation of the α-amine in EGFP over any of the ε-amines would correspond to an effective 189:1 selectivity for the α-amine over each of the ε-amines.

The focus of our study was to develop a general method for N-terminal chemical modification of proteins for use in chemical biology and in development of biopharmaceuticals. The chemical modification of the His tag with acetic acid derivatives is stable, in contrast to some other methods. The fact that ester **18** could introduce with high selectivity an azido moiety at 4 °C is a particularly attractive feature. We anticipate that the method also has the potential to be used for selective labeling of proteins in a cell lysate or even in cells for bioimaging.

## Methods

**Materials**. Synthesis and purification: *N,N′*-dicyclohexylcarbodiimide (DCC), dichloromethane (DCM), 4-(dimethylamino)pyridine (DMAP), heptane, ethyl acetate (EtOAc), *N,N*-dimethylformamide (DMF), benzyl bromoacetate, magnesium sulfate (MgSO$_4$), palladium hydroxide on carbon (Pd(OH)$_2$/C), Celite, triethylsilane (TES), diethyl ether (Et$_2$O), acetonitrile (MeCN), Sodium carbonate (Na$_2$CO$_3$), and gluconolactone (GDL) were purchased from Sigma-Aldrich (Brøndby, Denmark). *N*-[(1*H*-azabenzotriazol-1-yl)(dimethylamino)methylene]-*N*-methylmethanaminium hexafluorophosphate *N*-oxide (HATU), *N*-[(1*H*-benzotriazol-1-yl)(dimethylamino)methylene]-*N*-methylmethanaminium hexafluorophosphate *N*-oxide (HBTU), and 1-Hydroxy-7-azabenzotriazol (HOAt) were purchased from GL Biochem Ltd. (Shanghai, China). *N,N*

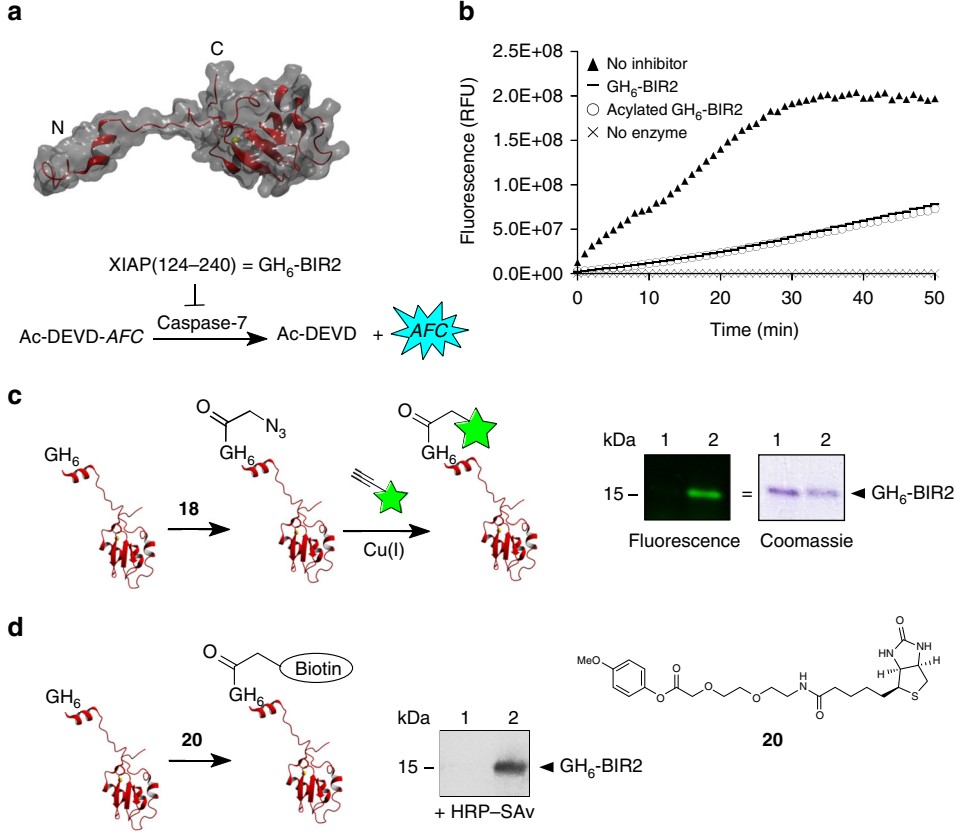

**Fig. 5** Functionalization of GH6-BIR2 with an azide, a fluorophore and biotin. **a** 3D structure of XIAP(124–240) with the linker region known to interact with caspase-3 and -7 depicted on the left and the BIR2 domain on the right. The reaction scheme of the inhibition assay is shown as well. The tetrapeptide DEVD is hydrolyzed by caspase-7 between the second aspartic acid and the AFC reporter group. AFC = 7-amino-4-trifluoromethylcoumarin. **b** Kinetic curves of the hydrolysis of Ac-DEVD-AFC by caspase-7, without inhibition (filled triangles) or when inhibited by unmodified (horizontal lines) or acylated (open circles) GH6-BIR2. The substrate only (crosses) was included as reference as well. **c** Two-step fluorescent labeling of GH6-BIR2 through acylation with 4-methoxyphenyl ester **18** followed by Cu(I)-catalyzed conjugation of alkyne cyanine dye 718, visualized by SDS-PAGE analysis. A fluorescence image and an image of the Coomassie-stained version of the same gel are shown. Lane 1: GH6-BIR2 treated with alkyne cyanine dye 718 and Cu(I) (negative control), lane 2: azido-functionalized GH6-BIR2 treated with alkyne cyanine dye 718 and Cu(I). **d** Direct biotinylation of GH6-BIR2 with 4-methoxyphenyl ester **20**, visualized by Western blot analysis. Binding of streptavidin to biotinylated GH6-BIR2 (lane 2) was detected by incubating the blot with HRP-SAv, a conjugate of streptavidin and horseradish peroxidase (HRP), followed by the addition of a chemoluminescent substrate of HRP. Unmodified GH6-BIR2 was loaded in lane 1 as negative control

'-diisopropylcarbodiimide (DIC), 1-methylimidazole (MeIm), N-methyl-2-pyrrolidone (NMP), N,N-dimethylformamide (DMF), N,N-diisopropylethylamine (DIPEA), piperidine, trifluoroacetic acid (TFA), and all Fmoc-protected amino acids were purchased from Iris Biotech GmbH (Marktredwitz, Germany). Sodium acetate (NaOAc) was acquired from Merck Millipore (Darmstadt, Germany). All peptide syntheses were carried out on Tenta Gel S Rink Amide (loading 0.22 mmol g$^{-1}$; Rapp Polymere GmbH). Amino acids were Fmoc protected at Nα-amino groups; side-chain protecting groups were tert-butyl (Ser, Tyr, and Glu), trityl (Trt, for Gln) and tert-butyloxycarbonyl (Boc, for Lys). Water was purified using an Ultra Clear Water System (Siemens) set at 0.055 μS cm$^{-1}$.

**Synthesis of phenyl 2-azidoacetates 16–18.** The phenol (200 mg) was dissolved in CH$_2$Cl$_2$ (5 mL), and 4-dimethylaminopyridine (0.1 equiv.) was added. 2-Azidoacetic acid (1.2 equiv.) was dissolved in CH$_2$Cl$_2$ (5 mL) with N,N-diisopropylcarbodiimide (1.2 equiv.) and stirred for 15 min, before it was added carefully to the phenol solution. The reaction mixture was stirred under N$_2$. After 2 h, additional 2-azidoacetic acid (1.2 equiv.) and N,N'-diisopropylcarbodiimide (0.6 equiv.) were added to the reaction mixture, and stirring was continued for 2 h. Then, the reaction mixture was concentrated by rotary evaporation, and the crude product was purified using an Isolera One instrument from Biotage™ equipped with a 25 g KP-Sil column. The product was eluted with CH$_2$Cl$_2$, and fractions containing the pure product were combined and concentrated by rotary evaporation. The product was dried in vacuo.

**Analysis of 4-nitrophenyl 2-azidoacetate (16).** Colorless oil, 294 mg (92%). $^1$H NMR (300 MHz, CDCl$_3$): δ 8.36–8.19 (m, 2H, 2 × H-Ar), 7.45–7.27 (m, 2H, 2 × H-Ar), 4.19 (s, 2H, CH$_2$). $^{13}$C NMR (75 MHz, CDCl$_3$) δ 166.18 (s, C = O), 154.67 (C-

Ar), 145.86 (C-Ar), 125.48 (2 × C-Ar), 122.27 (2 × C-Ar), 50.52 (CH$_2$). HR-MS (Q-TOF): m/z calcd. for chemical formula C$_8$H$_6$N$_4$O$_4$: 222.0389; found: [M + Na]$^+$ 245.0243 (Supplementary Fig. 23).

**Analysis of phenyl 2-azidoacetate (17).** Colorless oil, 294 mg (78%). $^1$H NMR (300 MHz, CDCl$_3$), δ 7.49–7.38 (m, 2H, 2 × H-Ar), 7.34–7.25 (m, 1H, H-Ar), 7.21–7.13 (m, 2H, 2 × H-Ar), 4.15 (s, 2H, CH$_2$). $^{13}$C NMR (75 MHz, CDCl$_3$) δ 166.95 (C = O), 150.23 (C-Ar), 129.71 (2 × C-Ar), 126.50 (C-Ar), 121.28 (2 × C-Ar), 50.54 (CH$_2$). HR-MS (Q-TOF): m/z calcd. for chemical formula C$_8$H$_7$N$_3$O$_2$: 177.0538; found: [M + Na]$^+$ 200.0426 (Supplementary Fig. 24).

**Analysis of 4-methoxyphenyl 2-azidoacetate (18).** Colorless oil, 316 mg (95%). $^1$H NMR (300 MHz, CDCl$_3$), δ 7.12–7.02 (m, 2H, 2 × H-Ar), 6.97–6.87 (m, 2H, 2 × H-Ar), 4.12 (s, 2H, CH$_2$), 3.82 (s, 3H, CH$_3$). $^{13}$C NMR (75 MHz, CDCl$_3$) δ 167.29 (C = O), 157.73 (C-Ar), 143.69 (C-Ar), 122.05 (2 × C-Ar), 114.67 (2 × C-Ar), 55.68 (CH$_3$), 50.49 (CH$_2$). HR-MS (Q-TOF): m/z calcd. for chemical formula C$_9$H$_9$N$_3$O$_3$: 207.0644; found: [M + Na]$^+$ 230.0503 (Supplementary Fig. 25).

Reagent **18** has the composition C$_9$H$_9$O$_3$N$_3$, which gives a (carbon + oxygen per azide nitrogen) ratio of 4. It can thus be considered relatively stable.

**Gram-scale synthesis of 4-methoxyphenyl 2-azidoacetate 18.** 4-Methoxyphenol (1.03 g, 8.3 mmol) and 4-dimethylaminopyridine (100 mg, 0.8 mmol) were dissolved in CH$_2$Cl$_2$ (50 mL) under stirring. 2-Azidoacetic acid (1.01 g, 10.0 mmol) was added. Then, N,N-diisopropylcarbodiimide (1.57 mL, 10.0 mmol) was added dropwise over a period of 10 min. The reaction mixture turned yellow to brown over 30 min. Stirring was continued for 1.5 h. Then, Celite (10 g) was added to the reaction mixture, and the solvent was removed by rotary

evaporation. The resulting brown Celite adsorbate was loaded on top of a pre-conditioned (ethyl acetate/heptane 1:39) column for vacuum liquid chromatography ($H \times D = 7 \times 5$ cm; silica gel 60 (0.015–0.040 mm)), and the product was eluted using a gradient of ethyl acetate/heptane (1:39→1:3). Fractions containing the pure product were combined and concentrated by rotary evaporation. The product was dried in vacuo to yield 1.63 g (95%) of **18** as a colorless oil.

**Synthesis of biotin OEG 4-methoxyphenyl ester 20.** A solution of 8-(9-fluor-enylmethyloxycarbonylamino)-3,6-dioxaoctanoic acid (1.93 g, 5 mmol, 5 equiv.) in dry DCM 4 mL) was added to 2-chlorotrityl chloride polystyrene resin (1 g, 1 mmol g$^{-1}$, 100–200 mesh, 1% DVB), and the reaction mixture was agitated for 16 h. The resin was washed with DCM ($5 \times 4$ mL) and DMF ($5 \times 4$ mL), and the Fmoc group was removed by treatment with 20% piperidine in DMF (4 mL) for 5 min., followed by 20% piperidine in DMF (4 mL) for 15 min. The resin was then washed with DMF ($5 \times 4$ mL), and DCM ($5 \times 4$ mL), followed by DMF ($5 \times 4$ mL).

Biotin (150 mg, 0.6 mmol) was preactivated with HATU (190 mg, 0.5 mmol), HOAt (75 mg, 0.55 mmol), and DIEA (150 µL, 0.85 mmol) in DMF (4 mL) for 5 min, and then added to the above resin (0.5 mmol). The reaction mixture was agitated for 2 h, and the resin was subsequently washed with DMF ($5 \times 4$ mL), followed by DCM ($5 \times 4$ mL). The resin was treated with TFA containing 5% water and 0.5% triethylsilane for 1 h. The cleaved biotin OEG acid was purified by RP-HPLC (on a Dionex Ultimate 3000 system) using a preparative C18 column (Phenomenex Gemini, 110 Å 5 µm C18 particles, $21 \times 100$ mm): Solvent A, water containing 0.1% TFA, and solvent B, acetonitrile containing 0.1% TFA, were used with gradient elution (0–5 min: 5–100% 5–32 min) at a flow rate of 15 mL min$^{-1}$. This material (55 mg, 0.14 mmol) was dissolved in dry DCM (5 mL), to which was added 4-methoxyphenol (20 mg, 0.16 mmol), 4-dimethylaminopyridine (2 mg, 0.1 mmol), followed by DIC (20 mg, 0.16 mmol). The reaction mixture was stirred for 16 h, after which it was concentrated by rotary evaporation. The product was purified by RP-HPLC (on a Dionex Ultimate 3000 system) using a preparative C18 column (Phenomenex Gemini, 110 Å 5 µm C18 particles, $21 \times 100$ mm): Solvent A, water containing 0.1% TFA, and solvent B, acetonitrile containing 0.1% TFA, were used with gradient elution (0–5 min: 5–100% 5–32 min) at a flow rate of 15 mL min$^{-1}$. The provided the product, [2-[2-[2-(D-biotinyl-amino)ethoxy]ethoxy]acetic acid 4-methoxyphenyl ester **20** (25 mg, 36%), as a white solid. $^1$H NMR (500 MHz, CD$_3$CN), $\delta$ 7.08–7.02 (m, 2H, $2 \times$ H-Ar), 6.97–6.92 (m, 2H, $2 \times$ H-Ar), 6.50 (br s, 1H, NH), 5.15 (br s, 1H, NH), 4.95 (br s, 1H, NH), 4.40 (ddt, $J = 1.0$ and 5.2 and 7.6 Hz, 1H, biotin CH), 4.37 (s, 2H, CH$_2$CO), 4.22 (ddd, $J = 2.0$ and 4.5 and 7.6 Hz, 1H, biotin CH), 3.79 (s, 3H, OMe), 3.74–3.71 (m, 2H, CH$_2$O), 3.63–3.60 (m, 2H, CH$_2$O), 3.49 (t, $J = 5.4$ Hz, 2 H, CH$_2$O), 3.30 (dd, $J = 5.6$ and 11.3 Hz, 2H, CH$_2$N), 3.17–3.11 (m, 1H, biotin CHS), 2.88 (dd, $J = 4.8$ and 12.7 Hz, 1H, biotin CH$_2$S), 2.63 (d, $J = 12.7$ Hz, 1H, CH$_2$S), 2.11 (t, partly overlapped by HDO, $J = 7.4$ Hz, 2H, biotin CH$_2$CO), 1.70–1.48 (m, 4H, biotin CH$_2$), 1.40–1.32 (m, 2H, biotin CH$_2$). $^{13}$C NMR (125 MHz, CD$_3$CN), $\delta$ 173.8, 170.8, 163.7, 158.5, 144.8, 123.5, 115.5, 71.6, 70.9, 70.4, 69.1, 62.3, 60.8, 56.4, 56.3, 41.2, 39.8, 36.3, 29.0, 28.9, 26.4. HR-MS (Q-TOF): $m/z$ calcd. for chemical formula C$_{23}$H$_{33}$N$_3$O$_7$S: 495.2039; found: [M + Na]$^+$ 518.1998; [M + H]$^+$ 496.2124 (Supplementary Fig. 26 and 27).

**Synthesis of peptides.** The Beltide-1 peptides were assembled by Fmoc solid-phase peptide synthesis on a Biotage$^®$ Initiator + Alstra$^{TM}$ microwave peptide synthesizer. The short peptides were synthesized on a parallel automated peptide synthesizer (Biotage$^®$ SYRO 2). The syntheses were carried out on Tenta Gel R Rink Amide resin (loading 0.22 mmol g$^{-1}$).

In the synthesis of the short peptides (Table 1), $N^\alpha$-Fmoc deprotection was performed at room temperature in two stages by treating the resin with piperidine/DMF (1:4) for 3 min followed by piperidine/DMF (1:3) for 15 min. The resin was then washed with NMP ($3\times$), DCM ($1\times$), and NMP ($3\times$). Peptide couplings were performed using 5.2 equiv. of Fmoc-AA in DMF (0.5 M), 5.2 equiv. of HOAt in DMF (0.5 M), 5.0 equiv. of HBTU in DMF (0.6 M), and 9.36 equiv. of DIEA in NMP (2 M). The coupling time was 120 min at RT, after which the resin was washed with NMP ($3\times$), DCM ($1\times$), and NMP ($3\times$).

In the synthesis of Beltide-1 peptides (Tables 2 and 3), $N^\alpha$-Fmoc deprotections were performed at room temperature in two stages by treating the resin with piperidine/DMF (1:4) for 3 min followed by piperidine/DMF (1:3) for 15 min. The resin was then washed with NMP ($3\times$). DCM ($1\times$). DMF ($1\times$). Peptide couplings were performed using 5.2 equiv. of Fmoc-AA in DMF (0.5 M), 5.2 equiv. of HOAt in DMF (0.5 M), 5.0 equiv. of HBTU in DMF (0.6 M) and 9.36 equiv. of DIEA in NMP (2 M). The coupling time was 10 min at 75 ℃, after which the resin was washed with NMP ($4\times$).

N-acetylated peptides were obtained by on-resin acetylation with Ac$_2$O in DMF. After the synthesis was completed the resin was washed with DCM ($6\times$) and thoroughly dried. The peptides were cleaved from the resin with TFA-H$_2$O-TES (95:3:2) for 5 min. and then for 2 h. The peptides were precipitated with cold diethyl ether to yield the crude products.

**Purification of peptides.** The peptides were purified by RP-HPLC (on a Dionex Ultimate 3000 system) with preparative C18 column (Phenomenex Gemini, 110 Å 5 µm C18 particles, $21 \times 100$ mm). For the short peptides (Table 1) solvent A, water containing 0.1% TFA; solvent B, acetonitrile containing 0.1% TFA, were used with gradient elution (0–5 min: 5–40% 5–32 min) at a flow rate of 15 mL min$^{-1}$, while for the Beltide containing peptides (Tables 2–3) solvent A, water containing 0.1% TFA, and solvent B, acetonitrile containing 0.1% TFA, were used with gradient elution (0–5 min: 5–100% 5–32 min) at a flow rate of 15 mL min$^{-1}$.

**Analysis of peptides and proteins.** Analyses of peptides were performed by UHPLC-MS on a RSLC Dionex Ultimate 3000 (Thermo) coupled to a QTOF Impact HD (Bruker) on a kinetex 2.6 µm EVO 100 Å C18 column ($50 \times 2.1$ mm, Phenomenex) which were used for the peptides and Aeris 3.6 µm widepore C4 column ($50 \times 2.1$ mm, Phenomenex) for the proteins. The following solvent system was used at a flow rate of 0.5 mL min$^{-1}$: solvent A, water containing 0.1% formic acid; solvent B, acetonitrile containing 0.1% formic acid. The column was eluted using a linear gradient from 5 to 100% of solvent B. Some short peptides (Table 1) were analyzed on analytical HPLC (Dionex Ultimate 3000) instrument with an analytical C18 column (Phenomenex Gemini NX 110 Å, 5 µm, C18 particles, $4.60 \times 50$ mm) coupled to a ESI-MS (MSQ Plus Mass Spectrometer, Thermo) using a linear gradient flow of water-acetonitrile containing 0.1% formic acid (Supplementary Tables 2 and 3). MALDI-TOF MS spectra were recorded on a Bruker autoflex$^{TM}$ speed MALDI-TOF instrument.

As for the protein modification reactions, conversions were based on relative abundance values calculated from deconvoluted MS data. Given the large number of positive charges that the proteins pick up when analyzed by ESI-MS, the effect of the loss of a single positive charge upon acylation of the N-terminus is expected to be negligible. EGFP picks up 20–40 charges, MBP 20 to >50, SUMO 8–20, and BIR-2 10–24.

**Expression of proteins.** The expression plasmid for GH$_6$-EGFP was generated from the pET15b-EGFP plasmid encoding GSSH$_6$-EGFP[44]. The two N-terminal serine codons were deleted by PCR using the following primers: 5′-TTTGTTTAACTTTAAGAAGGAGATATATACCATGGGCCATCATCATC-3′ and 5′-CTAGTTATTGCTCAGCGGT-3′. The insert, treated with NcoI and BamHI (New England Biolabs), was ligated in the pET15 vector, digested with the same restriction enzymes. DH5α cells (Invitrogen) were used for cloning. The plasmid was sequenced using the T7 forward primer (5′-TAATACGACTCAC TATAGGG-3′, Eurofins). E. coli BL21[DE3] cells were transformed with plasmid encoding GH$_6$-tagged EGFP or GSSH$_6$-tagged EGFP. Cultures were grown in 250 mL LB medium with 100 µg mL$^{-1}$ ampicillin and protein expression was induced with 1 mM IPTG for 4 h at 37 ℃. Cells were harvested and resuspended in 15 mL of 50 mM NaH$_2$PO$_4$, 300 mM NaCl at pH 8. Lysozyme was added to a final concentration of 1 mg mL$^{-1}$ and lysis was allowed to take place by incubation for 30 min on ice. The lysate was sonicated and centrifuged at $18{,}000 \times g$ at 4 ℃ for 30 min. The supernatant was incubated with 1 mL of Ni-NTA resin for 20 min under gentle stirring. The resin was loaded into a gravity column, washed with $2 \times$ 10 mL of lysis buffer (50 mM NaH$_2$PO$_4$, 300 mM NaCl at pH 8) and 10 mL of wash buffer (50 mM NaH$_2$PO$_4$, 300 mM NaCl, 2 mM imidazole at pH 8), and eluted in 3 mL elution buffer (50 mM NaH$_2$PO$_4$, 300 mM NaCl, 250 mM imidazole at pH 8). An additional purification step was performed on an ÄKTA$^{TM}$ pure system equipped with a HiLoad 16/600 Superdex 75 pg column (GE Healthcare). Fractions containing the protein (as confirmed by SDS-PAGE analysis) were pooled and concentrated using a centrifugal filter unit (Amicon, Ultra-15, MWCO 10 kDa). The protein concentration was determined by measuring absorbance at 490 nm on a Nanodrop 2000 (Thermo Scientific) using an extinction coefficient of 55,000 M$^{-1}$ cm$^{-1}$. The protein (4.4 mg mL$^{-1}$) was frozen in aliquots in liquid nitrogen and stored at $-20$ ℃.

GSH-tagged EGFP was produced by incubation of GSSH$_6$-tagged EGFP (100 µg) with thrombin (0.05 U, Thrombin cleavage capture kit, Novagen) for 16 h at room temperature. Efficient removal of the N-terminal peptide GSSHHHHHHSSGLVPR was confirmed by MALDI-TOF analysis. Biotinylated thrombin was removed by incubation with streptavidin agarose, according to the manufacturer's protocol.

The DNA coding for OneStrep-TEV-Gly-His-Sumo (smt3_Q12306) was ordered codon-optimized for E. coli from Geneart. Plasmid pNIC28-StrepTEVGlyHisSumo was created by digestion with NdeI and BamHI and ligation into the corresponding sites of plasmid pNIC28-Bsa4. After transformation of the plasmid into E. coli BL21[DE3], cells were cultured in 500 mL 2xYT medium containing 50 µg mL$^{-1}$ kanamycin at 37 ℃ until mid-exponential phase, at which time expression was induced with 1 mM IPTG for 16 h at 18 ℃. After harvesting, cells were lysed by three passes through an EmulsiFlex-C5 homogenizer (Avestin) at 10–15k psi and any debris and unbroken cells were removed by centrifuging at $18{,}000 \times g$ at 4 ℃ for 15 min. For protein purification, the supernatant was loaded onto a StrepTrap HP column (GE Healthcare, Piscataway, NJ, USA) on an Äkta$^{TM}$ pure system. Purification was performed as per the specifications of the manufacturer. Cleavage of the OneStrep tag was performed with 1:100 (w/w) recombinant His-tagged TEV protease (produced in-house) overnight at 4 ℃. TEV cleavage was assessed by SDS-PAGE and was found to be complete. TEV protease and SUMO protein with an N-terminal glycine residue followed by a His$_6$ tag were subsequently separated by size exclusion chromatography using a HiLoad 16/600 Superdex 75 pg column (GE Healthcare).

Competent E. coli BL21[DE3] cells were transfected with the pET28a plasmid containing the MBP coding sequence (malE_P0AEX9). They were grown in LB

medium at 37 °C with kanamycin and induced with 1 mM IPTG for 3 h. The cells were harvested by centrifugation at $4000 \times g$ for 10 min and resuspended in lysis buffer containing 40 mM Tris/HCl pH 8.5, 400 mM NaCl and 20 mM imidazole. The cells where lysed by two cycles of a French press at 35k psi. Cell debris was spun down for one hour at $18,000 \times g$. The supernatant was incubated with Ni-NTA resin for one hour under gentle stirring. The resin was loaded into a gravity column, washed in wash buffer containing 40 mM Tris/HCl pH 8.5, 1 M NaCl and 20 mM imidazole and eluted in elution buffer containing 40 mM Tris/HCl pH 8.5, 200 mM NaCl and 500 mM imidazole. DTT was added to a final concentration of 1 mM and cleavage of the N-terminal tag was performed with 1:100 (w/w) TEV protease. The cleavage reaction was left at 20 °C overnight. The protein was simultaneously dialyzed against a large reservoir of buffer containing 50 mM Tris/HCl pH 8.5. Minor precipitate was removed by centrifugation and the protein was loaded onto a 15 mL anion exchange column (Q-sepharose High Performance, GE Healthcare) equilibrated in buffer containing 50 mM Tris/HCl pH 8.5. A linear gradient of 5 column volumes using a buffer containing 50 mM Tris/HCl pH 8.5 and 1 M NaCl was used to elute the protein. Peak fractions were collected and dialyzed overnight at 4 °C into a buffer containing 200 mM HEPES pH 7.5 and 20 mM NaCl. Peak fractions were then concentrated, aliquoted, flash frozen, and stored at −80 °C until further use.

A gene fragment coding for residues 124–240 of human X-linked inhibitor of apoptosis protein (*xiap*_P98170) was ordered codon-optimized for *E. coli* from Eurofins. Plasmid pET15b-GH6-XIAP[124-240] was created by digestion with *Nco*I and *Bam*HI and ligation into the corresponding sites of plasmid pET15b. *E. coli* BL21[DE3] cells were transformed with the respective plasmid. A culture was grown in 750 mL LB medium with 100 µg mL⁻¹ ampicillin and protein expression was induced with 1 mM IPTG for 3.5 h at 37 °C. Cells were harvested and resuspended in 50 mM $NaH_2PO_4$, 300 mM NaCl at pH 8, supplemented with cOmplete™, EDTA-free protease inhibitor cocktail (Roche). Lysozyme was added to a final concentration of 1 mg mL⁻¹ and lysis was allowed to take place by incubation for 1 h on ice. The lysate was sonicated and centrifuged at $12,000 \times g$ at 4 °C for 45 min. Imidazole and 2-mercaptoethanol were added to the cleared lysate to a final concentration of 20 mM. The protein was purified over a HisTrap column (GE Healthcare), followed by a size-exclusion chromatography (SEC) step over a Superdex75 Increase 10/300 GL column (GE Healthcare) using 50 mM $NaH_2PO_4$, 150 mM NaCl at pH 7.5 as eluent. Purity was confirmed by SDS-PAGE. The fraction of endogenously gluconoylated protein (~25%) was efficiently removed by incubation of the purified protein at 37 °C for 1 day.

**Gluconoylation of model peptides.** A 2 mM solution of the corresponding peptide in 200 mM HEPES buffer at pH 7.5 was mixed with equal volume of 50 mM solution of GDL in 200 mM HEPES buffer at pH 7.5 and the resulting solution was incubated at room temperature. The progress of the reactions was monitored by LC–MS (Table 1).

**Gluconoylation of Beltides 13–15.** An aliquot of 100 µL of a freshly prepared 1 M solution of GDL in water was added to a 1 mL of 1 mM solution of Beltide derivative in 200 mM HEPES buffer at pH 7.5 at room temperature. The resulting solution was incubated at that temperature and the progress of the reaction was monitored by LC–MS (Table 2).

**Reaction of Beltides 13–15 with 4-methoxyphenyl ester 18.** An aliquot of 100 µL of 25 mM solution of 4-methoxyphenyl 2-azidoacetate **18** in acetonitrile was added to a 1 mL of 1 mM solution of Beltide derivative in 200 mM HEPES buffer at pH 7.5 at 4 °C. The resulting solution (2.5 mM in **18**) was incubated at that temperature and the progress of the reaction was monitored by LC–MS (Table 3 and Supplementary Table 3).

**Gluconoylation of proteins.** An aliquot of 10 µL of a freshly prepared 2 M solution of GDL in water was added to 100 µL of 35 µM solution of the corresponding EGFP variant in 200 mM HEPES buffer at pH 7.5 at room temperature. The resulting solution was incubated at that temperature and the progress of the reaction was monitored by LC–MS.

An aliquot of 10 µL of a freshly prepared 3.5 M solution of GDL in water was added to 100 µL of 350 µM solution of GH6-MBP in 200 mM HEPES buffer at pH 7.5 at room temperature. The resulting solution was incubated at that temperature and the progress of the reaction was monitored by LC–MS.

An aliquot of 10 µL of a freshly prepared 1.89 M solution of GDL in water was added to 100 µL of 42 µM solution of GH6-SUMO in 200 mM HEPES buffer at pH 7.5 at room temperature. The resulting solution was incubated at that temperature and the progress of the reaction was monitored by LC–MS.

**Reaction of GH6 proteins with 4-methoxyphenyl ester 18.** An aliquot of 23 µL of a 14 mM solution of 4-methoxyphenyl 2-azidoacetate **18** in acetonitrile was added to 80 µL of a 35 µM solution of the GH6-GFP in 200 mM HEPES buffer at pH 7.5 at 4 °C. The resulting solution was incubated at that temperature and the progress of the reaction was monitored by LC–MS.

An aliquot of 2 µL of a 14 M solution of 4-methoxyphenyl 2-azidoacetate **18** in acetonitrile was added to 20 µL of a 35 µM solution of the GSH-EGFP and

GSSH6-EGFP variants in 200 mM HEPES buffer at pH 7.5 at 4 °C. The resulting solution was incubated at that temperature and the progress of the reaction was monitored by LC–MS.

An aliquot of 8 µL of a 14 mM solution of 4-methoxyphenyl 2-azidoacetate **18** in acetonitrile was added to 80 µL of a 35 µM solution of GH6-MBP in 200 mM HEPES buffer at pH 7.5 at 4 °C. The resulting solution was incubated at that temperature and the progress of the reaction was monitored by LC–MS.

An aliquot of 8 µL of a 31 mM solution of 4-methoxyphenyl 2-azidoacetate **18** in acetonitrile was added to 80 µL of a 170 µM solution of GH6-SUMO in 200 mM HEPES buffer at pH 7.5 at 4 °C. The resulting solution was incubated at that temperature for a period of 16 h, and the progress of the reaction was monitored by LC–MS. Different reaction conditions were investigated in order to optimize the conversion and lower side reactions; the 4-methoxyphenyl 2-azidoacetate **18** concentration (from 2.5 to 40 equiv.), the temperature (4 or 25 °C), and the reaction time (1–36 h) were varied (Supplementary Table 1).

An aliquot of 2 µL of a stock solution of 4-methoxyphenyl 2-azidoacetate **18** in acetonitrile was added to 30 µL of a 40 µM solution of GH6-Bir2 in 50 mM $NaH_2PO_4$, 150 mM NaCl at pH 7.5 at 4 °C. (NB: Here, the acylation reaction was performed in the buffer used during the final purification step of GH6-BIR2. It underscores that the reaction is not limited to HEPES buffer and that it can be performed in other (more common) buffers, too.) The concentration of **18** in the stock solution was adjusted according to the number of equiv. that was to be assessed (ranging from 7.5 to 40 equiv.). The reactions were incubated at 4 °C for 16–20 h and analyzed by LC–MS.

**Two-step biotinylation and PEGylation of GH6-EGFP.** An aliquot of 23 µL of a 14 mM solution of 4-methoxyphenyl 2-azidoacetate **18** in acetonitrile was added to 80 µL of a 35 µM solution of GH6-EGFP variant in 200 mM HEPES buffer at pH 7.5 at 4 °C. The resulting solution was incubated at that temperature and the progress of the reaction was monitored by LC–MS. After 3 days excess of reagent and/or hydrolysis products were removed by repetitive (5×) centrifugal filtration (Millipore Amicon Centriplus 3 kDa cut-off) and redissolution (10× dilution) steps using 100 mM sodium phosphate buffer at pH 7.5 as washing solution to obtain 80 µL of 100 µM solution of azido-functionalized $N_3$-$CH_2$-C(O)-GH6-EGFP in 100 mM sodium phosphate buffer at pH 7.5 that was used for subsequent conjugation with DBCO-PEG4-Biotin **19** and 5 kDa DBCO-PEG.

An aliquot of 1 µL of a 6 mM solution of DBCO-PEG4-Biotin **19** in $CH_3CN$/$H_2O$ 1:1 was added to 40 µL of a 100 µM azido-functionalized $N_3$-$CH_2$-C(O)-GH6-EGFP in 100 mM sodium phosphate buffer at pH 7.5 and room temperature. The resulting solution was incubated at that temperature and the progress of the reaction was followed by LC–MS.

An aliquot of 1 µL of a 2.5 mM solution of 5 kDa DBCO-PEG in water was added to 20 µL of a 100 µM azido-functionalized $N_3$-$CH_2$-C(O)-GH6-EGFP in 100 mM sodium phosphate buffer at pH 7.5 and room temperature. The resulting solution was incubated at that temperature and the progress of the reaction was followed by LC–MS.

**Fluorescent labeling of GH6-BIR2.** Thirty equivalent of alkyne cyanine 718 (1.5 µL of a 10 mM stock solution in DMSO, Sigma Aldrich) were added to 10 µL of 40 µM GH6-BIR2 pre-modified with **18**. An aliquot of 1.2 µL of a freshly prepared mixture of 2.5 mM $CuSO_4$, 12.5 mM Tris(3-hydroxy propyltriazolylmethyl)amine (THPTA) and 37.5 mM sodium ascorbate was added to catalyze the cycloaddition. As a negative control, non-modified GH6-BIR2 was treated with alkyne fluorophore and Cu(I) catalyst as well. The reactions were incubated at room temperature overnight while kept in the dark, and analyzed by SDS-PAGE. Fluorescent bands were detected on a Typhoon 7000 FLA laser scanner ($\lambda_{ex}$ 635 nm, $\lambda_{em}$ 670 nm, PMT at 1000). Afterwards, the gel was stained with Coomassie Brilliant Blue R250.

**Caspase-7 inhibition assay.** Caspase-7 (500 nM) was pre-activated by incubation for 15 min at 37 °C in assay buffer (20 mM PIPES (pH 7.4), 100 mM NaCl, 1 mM EDTA, 0.1% CHAPS, 10% sucrose and 10 mM DTT). Then, BIR2 was added in equimolar amount to the activated enzyme, followed by an additional incubation step for 15 min at 37 °C. The enzyme/inhibitor solution (50 µL) was mixed with a pre-heated solution of Ac-DEVD-AFC (50 µL, 200 µM in assay buffer, 2% DMSO), resulting in final concentrations of 100 µM substrate and 250 nM caspase-7/BIR2. The formation of fluorescent product was followed at 37 °C for 1 h using a SpectraMax i3x plate reader ($\lambda_{ex}$ = 405 nm, $\lambda_{em}$ = 500 nm). Measurements were performed in duplicate.

**Biotinylation of GH6-BIR2.** Twenty equivalent of compound **20** (4 µL of a 10 mM stock solution in 1:1 acetonitrile:water) were added to 50 µL of 40 µM GH6-BIR2 in 50 mM $NaH_2PO_4$, 150 mM NaCl (pH 7.5). The reaction was incubated at 4 °C for 24 h and analyzed by LC–MS and western blot.

**Trypsin and chymotrypsin digestion.** Excess of GDL, phenyl ester **18** or DBCO-PEG4-biotin were removed by repeated (five times) centrifugal filtration using a 0.5-mL Amicon Ultra centrifugal filter (10 MCWO). Trypsin (sequencing-grade, modified, Sigma-Aldrich) digestions were performed overnight at 37 °C in 0.1 M

$NH_4HCO_3$ (pH 8). Chymotrypsin (sequencing grade, Sigma-Aldrich) digestions were performed overnight at room temperature in 0.1 M $NH_4HCO_3$ (pH 8). Digests were analyzed by MALDI-TOF and LC–MS.

**SDS-PAGE analysis**. Proteins (2–5 µg) were mixed with 4× Laemmli sample buffer supplemented with 2-mercaptoethanol (BioRad), boiled at 95 °C for 2–5 min and loaded on an any kD™ Mini-PROTEAN® TGX™ precast protein gel (Biorad). The Dual Color Precision Plus Protein™ Standard (BioRad) was used as reference. The gel was run at 200 V, stained in 0.1% (w/v) Coomassie Brilliant Blue R250 solution (10% (v/v) AcOH, 50% (v/v) MeOH in MilliQ water) and destained in destaining solution (10% (v/v) AcOH, 50% (v/v) MeOH in MilliQ water) (see Supplementary Fig. 28 for uncropped versions of the gels).

**Data availability**. All relevant data are available from the authors.

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

## Acknowledgements

The VILLUM FONDEN is acknowledged for funding the Biomolecular Nanoscale Engineering Center (BioNEC), a VILLUM center of excellence, grant number VKR022710, and an HPLC-MS instrument (VKR9912 to K.J.J.). A DFF-Mobilex grant from DFF and the European Commission (4093-00083 to S.S.) is acknowledged. The Novo Nordisk Fonden is acknowledged for financial support (S.K). Helle Munck Petersen is acknowledged for technical assistance. Amine Arslan is acknowledged for expression and purification of caspase-7.

## Author contributions

The present work was designed and supervised by K.J.J. K.K.S. and C.T.H. synthesized and purified all peptides, while C.T.H. performed experiments on the gluconoylation of short peptides (Table 1). J.E.R. did additional experiments on gluconoylation and on oxidation. S.K. expressed and purified the SUMO variant, S.R.M. expressed and purified the MBP variant, while S.S. expressed and purified the EGFP and BIR-2 variants. K.V. and M.B.T. synthesized phenyl esters. M.C.M.-M. performed experiments on beltide-1 peptides (Tables 2 and 3), SUMO, EGFP variants, and MBP. S.S. supervised the protein chemistry and performed experiments on acylation of EGFP, MBP, and BIR-2. K.K.S. and S.S. performed the mass spectrometry. M.B.T. performed the mechanistic studies on model compounds. K.K.S. and M.B.T. performed further optimization experiments on

SUMO. K.J.J., M.C.M.-M., and S.S. cowrote the draft manuscript. All authors edited the manuscript.

## Additional information

**Competing interests:** The authors declare no competing interests.

