## [Peer Review File · Nature Communications]

Reviewers' comments:

Reviewer #1 (Remarks to the Author):

Martos-Maldonado et al. report the selective addition of an acyl group to the N-terminal alpha-amino group of a glycine on different peptides and proteins by treatment with 4-methoxyphenyl esters as acyl donors. Selectivity is mediated by a neighboring histidine tag of preferably 6 histidine residues that acts as an imidazole acylation catalyst (base catalysis). The authors convincingly describe how they have based the development of their approach on the generally known fact that proteins with N-terminal histags can get modified by gluconic acid lactone during expression in *E. coli*. This modification has been described in the late 1990 and is probably related to an accumulation of the lactone due to miss-regulation in the bacterial sugar metabolism. Here, a straightforward series of experiments using peptides and proteins with different N-terminal amino acids and histags has been analyzed with the result that an N-terminal glycine residue followed by a hexahistidine tag gives the best yields and highest selectivity when treated with a large excess (100 eq.) of the lactone. No double modified peptides are reported and controls such as N-terminal acetylation that completely blocks modification by the lactone are included as well. Also protein modification with high conversion was observed.

Finding even better suited acylating reagents, namely p-methoxyphenyl esters, are the second highlight of this manuscript. With these, peptides such as Beltide-1 can be functionally modified with only 2.5 equivalents, e.g. by introducing an azide for further reactions. However, there are some drawbacks, for example the occurrence of di- and tri-acylated byproducts, which have not been detected with the lactone. In figure legend 3 a short reference to the "partial reversibility" of the acylation reaction is made. This point is never explained or discussed in the main text. But figure SI-6 shows a very significant loss of the gluconoylation of peptide 15. This needs to be explained!

In addition, the reaction with proteins such as EGFP, MBP and SUMO are much slower and require a 60 fold excess of the p-methoxyphenyl ester reagent over 6 days. The long reaction time is only feasible when very stable proteins are targeted as more sensitive ones will degrade. Shorter reaction times reduce the modification efficiency considerably. Subsequent modification of azide groups within the transferred acyl group give more complex modification patterns but direct attachment of larger groups activated as p-methoxyphenyl esters only gave moderate yields after 2 days.

Overall, I believe that this strategy is of very high interest to all researches looking for selective protein modification strategies. Demand for these is very high to generate samples for biochemical, cellular and therapeutic applications (as stated in the manuscript). In addition, all proteins already expressed with an N-terminal his-tag are amenable to this modification scheme (or can be adapted by simple cloning steps). In the discussion section the authors present a theoretical basis for the observed selectivity, mainly based on proximity of the N-terminus to the imidazole side chains and their variable pKa values. These values fit well to the properties of the p-methoxyphenyl esters used here. The potential use of this approach and the high compatibility with physiological conditions are strong arguments for publication Nature Chemistry.

However, some points raised above and listed below need to be addressed:

- Quantification of most modification reactions reported here are based on MS data. The authors need to show/argue that this is reliable and not significantly influenced by the fact that the N-terminal acylation removes a positive charge from the analyte.
- How does a lysine residue just C-terminal of the hexahis tag react? Here, proximity to the imidazole side chains should make acylation much more likely.
- Where does the di- and triacylation (table 3) occur?
- The authors briefly mention the benefits of a "free" alpha-amino group when identifying glycine

as the best N-terminal residue. They should elaborate on this argument. How does a small side chain interfere with acylation/imidazole catalysis? And how does an N-terminal histidine perform? Only peptide 6 in table 2 (caption missing on page 5) gives a hint that His could also work well.

- Did the authors ever check if the addition of imidazole or a short hexahistidine peptide to their labeling reactions could also mediate the acylation reaction in the absence of the histag or would lead to less selective modification?

- The authors should add a reference to the work by Brecher and Balls with p-nitrophenyl acetate in the presence of imidazoles (e.g. JBC 1957, 227, 845-51).

- p. 12, line 6: "specie" should be "species"

Reviewer #2 (Remarks to the Author):

This manuscript from the Jensen lab reported a new N-terminal modification strategy based on known observation of His-tag modifications. The optimized sequence "GHHHHHH" serves as a nucleophile for mild modifications. Furthermore, electrophiles with bio-orthogonal functional groups can be introduced easily on the N-termini. Overall, this is an elegant manuscript with convincing data to support the conclusion. The following experiments may make this manuscript even stronger.

1. There is no stability data about the modified peptides and proteins in this manuscript. Whether the goal is to create research tools or biopharmaceuticals, a stable linkage is required. Given the special basic microenvironment created by the His tag, I would like to see the stability of the amide linkage in buffers with various pH or serum.

2. I agree that His6 is the most common length people used for purification. However, sometimes, more than one His6 tags were inserted in proteins for better purification results. It will be nice if the authors can provide an example of a protein with at least 2 His6 tags to demonstrate the scope.

3. The authors mentioned that longer time (24hrs) is required for the ester 18 while only 1-2 hr is required for GDL as substrates. This means that a potential GDL analog with bio-orthogonal functional groups may be even better substrate for modifications. I wonder if the authors can discuss further about this (why longer time needed for ester 18 and why not go for GDL analog?).

4. While the authors gave a brief introduction of existing N-terminal modification methods, the following papers should be recognized and cited due to the shared scope and purpose with the current manuscript.

J. Am. Chem. Soc., 2006, 128 (46), pp 14796-14797

Chem. Commun., 2013, 49, 6888-6890

Chem. Sci., 2017, 8, 2717-2722

Reviewer #3 (Remarks to the Author):

The groups of Schoffelen and Jensen report in their paper on Selective N-terminal acylation of peptides and proteins with an optimized His sequence by 4-methoxyphenyl esters. It is in the belief of this referee that this article does not warrant a publication in a high impact factor as Nat Comm. because of the following reasons:

1) The majority of the work presented in this paper (experiments in table 1 and 2 and figure 3) deals with N-terminal gluconoylation of expressed proteins and synthetic peptides bearing His tag

using D-Gluconic acid δ -lactone (GDL). Nevertheless, this reactivity has been already reported by Geoghegan et al., with the N-terminal phosphogluconoylation of fused proteins and synthetic peptides in *Anal biochem.* 1999, 267, 169.

2) For the N-terminal acylation of the His tag using the ester 18 or 20, the selectivity of these reactions is controversial regarding to the statement of the authors: at protein level poly-acylated conjugates are always observed cf fig 4 B and SI-9, and fig 4 G, for 18 and 20 respectively. Furthermore, even with lower amount of compound 18 (10-30 equiv.) the selectivity is poor cf fig SI-7. On top of that, there is a huge lack of consistency in the conditions used for protein modifications and in some case there is mismatch between the conditions written in manuscript and the SI e.g. reaction conditions in table 3 p 7 different from those given in the experimental section p21

3) No explanation given for some reactivity: what is the rationale behind the difference in reactivity between 15 and 15G? Why the percentage of poly-acylated products decrease when going from 15 and 15G?

4) What about the stability of the ester 18 in the reaction conditions? This is not discussed at all.

5) Finally, the attempt to rationalize the reactivity with the control experiments is useless. Indeed, comparing the reactivities of acetic anhydride and ester 18 with imidazole is like comparing carrot and cabbages. Even though compound 18 has a borderline reactivity the choice of acetic anhydride which is structurally and from a reactivity perspective unrelated is not appropriate: the difference in reactivity: electrophilicity and steric hindrance between these two derivatives is non-comparable and therefore, the conclusion made for the mechanism is not relevant.

6) Although the authors have applied their acylation reaction to different protein substrates I lack the report of a convincing application of the reported protocol, which should be included in a journal like *Nature Comm.*

Minor points:

The NMR spectra of synthesized derivatives should be included, as well as the full characterization spectra of the conjugates.

In the experimental part: The synthetic protocol of compound 16-18 (p16) is not clear and can be confusing.

For synthesis of the peptides as well (p17), the protocol has to be written again, it is not clear and can be misleading.

Some typos e.g. p6, numeration of the figure should be coherent: Fig SI-4 appears in the text before Fig SI-2 and p7 acylation conditions of 13AC not provided.

We appreciated the constructive and helpful comments by reviewers 1 and 2. In the following, we have very carefully replied to all comments and questions by all three reviewers. We have performed a number of experiments to answer some key questions.

Reviewer #1

The reviewer writes

“Overall, I believe that this strategy is of very high interest to all researches looking for selective protein modification strategies. Demand for these is very high to generate samples for biochemical, cellular and therapeutic applications (as stated in the manuscript). In addition, all proteins already expressed with an N-terminal his-tag are amenable to this modification scheme (or can be adapted by simple cloning steps).

The potential use of this approach and the high compatibility with physiological conditions are strong arguments for publication Nature Chemistry.

1. However, there are some drawbacks, for example the occurrence of di- and tri-acylated byproducts, which have not been detected with the lactone. In figure legend 3 a short reference to the “**partial reversibility**” of the acylation reaction is made. This point is never explained or discussed in the main text. But figure SI-6 shows a very significant loss of the gluconoylation of peptide 15. This needs to be explained!”

We agree with the reviewer that the reversibility of the gluconoylation reaction is interesting. In the original submission we included a figure (new Fig SI-9), which clearly showed this for a gluconoylated peptide. In response to this question by the reviewer, we studied the reversibility of the gluconoylation of GH₆-EGFP. This new study also showed reversibility, hence that a gluconoylated protein can be degluconoylated. Also in response to this and another question, we studied the stability of the peptides and proteins acylated with our 4-methoxyphenyl ester reagents. This acylation was stable (Fig SI-10 and SI-11).

The reversibility of the gluconoylation could be due to the polyol structure. However, we find that a detailed study of this is beyond the scope of this proof-of-concept paper.

2A.

“In addition, the reaction with proteins such as EGFP, MBP and SUMO are much slower and require a 60 fold excess of the p-methoxyphenyl ester reagent over 6 days. The long reaction time is only feasible when very stable proteins are targeted as more sensitive ones will degrade. Shorter reaction times reduce the modification efficiency considerably.”

The reaction times reported in the original submission were due to practical considerations. We appreciate the suggestion to shorten the reaction times. We went back to lab and studied the acylation of EGFP at 4 °C for 1-2 days. It was observed that no additional modification occurred after one day unless a fresh portion of the 4-methoxyphenyl ester was added. We re-assessed the effect of adding fresh portions of the acylating agent in consecutive days. We have included one of the results from this new study in figure SI-12.

In addition, we have clarified the text describing the reaction conditions.

2B

“Subsequent modification of azide groups within the transferred acyl group give more complex modification patterns but direct attachment of larger groups activated as p-methoxyphenyl esters only gave moderate yields after 2 days.”

We are not quite sure what the reviewer means by ‘more complex’. Perhaps the reviewer means that the peaks of the starting material (SM) and P2 in Fig 4C are broader and less smooth than the ‘SM’ and ‘P2’ peaks in Fig 4B. We have re-analyzed our MS data and it seems that this ‘peak broadening’ is an artefact of the peak picking and deconvolution procedure.

*In case the reviewer was thinking of the selectivity question, then we would like to emphasize that the reaction with **18** and **20** still is highly selective (and with fewer equivalents than for gluconolactone), introduces very useful functionalities and likely can be tuned further.*

3

“Quantification of most modification reactions reported here are based on MS data. The authors need to show/argue that this is reliable and not significantly influenced by the fact that the N-terminal acylation removes a positive charge from the analyte.”

In general, peptides were quantified by HPLC UV-absorption (with one exception, Fig SI-8). We have now explicitly stated this in the table headings in the main article. We have added a note to explain the quantification of ratios between closely related proteins by mass spectrometry.

4

“How does a lysine residue just C-terminal of the hexahis tag react? Here, proximity to the imidazole side chains should make acylation much more likely.”

*We appreciated this question and went back to lab to answer it. We synthesized two new peptides, **15K** and **15K-Ac**, to correctly answer this. We saw a somewhat elevated level of acylation of the Lys side-chain amine that was placed directly C-terminal of our His acylation tag. However, N-terminal acylation of the Gly was still clearly preferred. This confirmed our concept. These new experiments are a good extension of our study and we thank the reviewer for the suggestion. The data is in the updated Table 3.*

5

“Where does the di- and triacylation (table 3) occur?”

*We performed a trypsin digestion on di-acylated **15Az-Az**, followed by mass spectrometry, and could confirm that it occurs mainly on Lys-4 (new Fig SI-7).*

We observed very little if any tri-acylated products, and have updated the main article.

6

“The authors briefly mention the benefits of a “free” alpha-amino group when identifying glycine as the best N-terminal residue. They should elaborate on this argument. **How** does a small side chain interfere with acylation/imidazole catalysis? And how does an N-terminal histidine perform?”

Only peptide 6 in table 2 (caption missing on page 5) gives a hint that His could also work well.

We find that the data in Table 1 clearly show that an N-terminal Gly is highly preferred over amino acids with more bulky side-chains, such as Ala and Val (10 vs 11 and 12). Also, a Gly is preferred over His (10 vs 6; 7 vs 5).

7

“Did the authors ever check if the addition of imidazole or a short hexahistidine peptide to their labeling reactions could also mediate the acylation reaction in the absence of the his-tag or would lead to less selective modification?”

We appreciated this interesting suggestion and performed an additional experiment. GSH-EGFP was treated with the acylation reagent 18 in the presence of Ac-His₆-NH₂ (new Fig SI-15). The Ac-His₆-NH₂ did not catalyze the reaction, which confirmed the need for a covalent, N-terminal His acylation tag to autocatalyze the acylation.

8

“The authors should add a reference to the work by Brecher and Balls with p-nitrophenyl acetate in the presence of imidazoles (e.g. JBC 1957, 227, 845-51).”

We have done so.

9

“p. 12, line 6: “specie” should be “species”

We have corrected this.

Reviewer #2

“Overall, this is an elegant manuscript with convincing data to support the conclusion. The following experiments may make this manuscript even stronger.

1. There is no stability data about the modified peptides and proteins in this manuscript. Whether the goal is to create research tools or biopharmaceuticals, a stable linkage is required. Given the special basic microenvironment create by the His tag, I would like to see the stability of the amide linkage in buffers with various pH or serum.”

As mentioned above, we have performed additional experiments, which show that peptides and proteins acylated with 4-methoxyphenyl esters form stable, acylated products (new Fig SI-10 and Fig SI-11).

2.

“I agree that His₆ is the most common length people used for purification. However, sometimes, more than one His₆ tags were inserted in proteins for better purification results. It will be nice if the authors can provide an example of a protein with at least 2 His₆ tags to demonstrate the scope.”

We focused on the developing the His acylation tag with the shortest His segment that would also allow reliable purification by Ni²⁺ NTA affinity chromatography. We see no reasons why our His acylation tag method should not work also with longer His sequences.

3.

“The authors mentioned that longer time (24hrs) is required for the ester 18 while only 1-2 hr is required for GDL as substrates. This means that a potential GDL analog with bio-orthogonal functional groups may be even better substrate for modifications. I wonder if the authors can discuss further about this (why longer time needed for ester 18 and why not go for GDL analog?).”

This is an interesting question. First, it is important to notice that gluconolactone (Gdl) is used in vast excess. Gdl is commercially available and rather inexpensive. Any Gdl would have to be chemically synthesized, which would make it expensive and thus unattractive. We actually did try to synthesize a Gdl derivative, which proved very difficult. We were therefore pleased that we could develop 4-methoxyphenyl esters as relatively inexpensive reagents that required less excess than Gdl for acylation.

4.

“While the authors gave a brief introduction of existing N-terminal modification methods, the following papers should be recognized and cited due to the shared scope and purpose with the current manuscript.

J. Am. Chem. Soc., 2006, 128 (46), pp 14796–14797

Chem. Commun., 2013, 49, 6888-6890

Chem. Sci., 2017, 8, 2717-2722”

We have added these references.

Reviewer #3

1

“The majority of the work presented in this paper (experiments in table 1 and 2 and figure 3) deals with N-terminal gluconoylation of expressed proteins and synthetic peptides bearing His tag using D-Gluconic acid δ -lactone (GDL). Nevertheless, this reactivity has been already reported by Geoghegan et al., with the N-terminal phosphogluconoylation of fused proteins and synthetic peptides in Anal biochem. 1999, 267, 169.”

We clearly refer to the work of Geoghegan and coworkers in the Introduction. The reviewer seems to overlook, that the gluconoylation is only the first part of our paper and that the second part describes the development of improved and truly useful reagents. Secondly, the reviewer appears to overlook that we optimize the reported side-reaction by ‘tuning’ the His tag sequence to make a His acylation tag that is different from conventional His tags.

2A

“For the N-terminal acylation of the His tag using the ester 18 or 20, the selectivity of these reactions is controversial regarding to the statement of the authors: at protein level poly-acylated conjugates are always observed cf fig 4 B and SI-9, and fig 4 G, for 18 and 20 respectively. Furthermore, even with lower amount of compound 18 (10-30 equiv.) the selectivity is poor cf fig SI-7.”

While the reagents 18 and 20 are far more useful than Gdl, they are also somewhat less selective than Gdl. However, Gdl requires a very large excess. The selectivity for N-terminal acylation over Lys acylation by 18 and 20 is still remarkably high and provides a very useful approach to N-terminal acylation.

2B

“On top of that, there is a huge lack of consistency in the conditions used for protein modifications and in some case there is mismatch between the conditions written in manuscript and the SI e.g. reaction conditions in table 3 p 7 different from those given in the experimental section p21”

We do not see any ‘huge lack of consistency’ nor a discrepancy between Table 3 and the Experimental on p 21. However, we have added a short sentence in the experimental section to clarify the final concentration of phenyl ester 18.

3)

“No explanation given for some reactivity: what is the rationale behind the difference in reactivity between 15 and 15G? Why the percentage of poly-acylated products decrease when going from 15 and 15G?”

We have redone some of the experiments in Table 3 and have added results for new peptides (15K and 15K-Ac). There is a minor but reproducible difference between 15 and 15G. We find that it is not of great importance.

4)

“What about the stability of the ester 18 in the reaction conditions? This is not discussed at all.”

We appreciate the suggestion. We have studied the half-life and report it in the revised manuscript (new Fig SI-4).

5)

“Finally, the attempt to rationalize the reactivity with the control experiments is useless. Indeed, comparing the reactivities of acetic anhydride and ester 18 with imidazole is like comparing carrot and cabbages. Even though compound 18 has a borderline reactivity the choice of acetic anhydride which is structurally and from a reactivity perspective unrelated is not appropriate: the difference in reactivity: electrophilicity and steric hindrance between these two derivatives is non-comparable and therefore, the conclusion made for the mechanism is not relevant.”

*We believe the reviewer did not fully understand the design of the mechanistic studies. We are not comparing the reactivity of 18 and acetic anhydride. Rather, control experiments by NMR (SI-16) and UV spectroscopy (SI-17), show the absence of covalent N-acyl imidazole derivatives when 18 is used as acylating agent. Acetic anhydride is included as a **positive control** in both cases to indicate the location of absorptions upon N-acylation. A small error (introduced at a late stage of editing) on page 13 could be the reason for the confusion. We have changed the wording from: "Furthermore, we compared hydrolysis rates of 18 and Ac₂O in the presence of imidazole or 2-isopropylimidazole..." to "Furthermore, we compared hydrolysis rates of 18 in the presence of imidazole or 2-isopropylimidazole..."*

Minor points

"The NMR spectra of synthesized derivatives should be included, as well as the full characterization spectra of the conjugates."

We have now included the spectra. We already had NMR spectra for conjugate 9.

"In the experimental part: The synthetic protocol of compound 16-18 (p16) is not clear and can be confusing."

We have revised the protocol for the synthesis of 18.

"For synthesis of the peptides as well (p17), the protocol has to be written again, it is not clear and can be misleading."

We have slightly revised the section on the synthesis of the peptides.

"Some typos e.g. p6, numeration of the figure should be coherent: Fig SI-4 appears in the text before Fig SI-2 and p7 acylation conditions of 13AC not provided."

Conditions for acetylation of the peptides have been added to the experimental section.

Additional editing

In our own editing, we removed the experimental on the effect of gluconolactone hydrolysis on pH, as there was no reference to this in the main article.

Reviewers' comments:

Reviewer #1 (Remarks to the Author):

Martos-Maldonado et al. have addressed most of the concerns voiced by reviewers 1 and 2 during the first round of reviewing making the manuscript much stronger. The additional stability data, control reactions and more detailed reactions conditions will help to convince the many potential users of this N-terminal labeling approach for His-tagged proteins.

All relevant data has been included in the main manuscript and the supporting information.

Therefore, I now recommend publication of the manuscript in Nature Communications.

Reviewer #2 (Remarks to the Author):

The authors have done a comprehensive job to improve the manuscript and address my previous comments. Therefore, I am supportive of accepting this manuscript.

Reviewer #3 (Remarks to the Author):

Referee report

The groups of Schoffelen and Jensen report on selective N-terminal acylations of peptides and proteins with an optimized His-sequence using D-Glucono-1,5-lactone (GDL) and 4-methoxyphenyl esters. Overall, these are interesting findings and should be published in a bioorganic chemistry journal; however, it is in the referee's belief that this article does not warrant a publication in a high-impact general science journal like Nature Communications for the following reasons:

1. I want to emphasize again that this approach is an incremental work and not conceptually new. The selective N-terminus phosphogluconoylation of expressed proteins containing hexa histidine "His Tag" has been already disclosed (cf Geoghegan et al. *Anal biochem.* 1999, 267, 169). The authors also reported the gluconoylation of different synthetic peptides (i.e. GSSHHHHHSSGLVPR GSAHHHHAAR, GASHHHHAAR and GAAHHHHAAR) containing hexa histidine and tetra histidine residues using the commercially available D-Glucono-1,5-lactone (GDL) and in good yields (70%). On top of that, the authors carefully studied this reaction and provided significant and solid insights into the mechanism using different techniques (MS, NMR...). Finally, they rationalized this transformation through base catalysis or nucleophilic catalysis mechanism, with acyl transfer from histidine. Taken together, the presented paper is based on established work and an already reported concept.

2. Despite the fact that the authors took advantage from this known reaction to create another acylation reagent (in the manuscript the 4-methoxyphenyl ester 18) the scope of the presented approach is rather narrow, as the authors limit themselves to only very few derivatives of 4-methoxyphenyl esters, namely containing an azide in 18 or a triazole substituent after reaction of 18 with a strained alkyne. For a paper aiming to introduce a new protein modification method it would be essential to know if the scope of 4-methoxyphenyl ester derivatives is limited to these derivatives and in light of the lack of novelty of the overall concept (see 1.), I think that the current manuscript should address this.

3. Closely related to 2., I expect that in a high-impact paper like Nature Communication a convincing application of a protein is given in which the functionality of the modified protein is critically addressed or the benefit of the particular modification strategy becomes visible. The

acylation of therapeutically or biologically relevant proteins followed by their chemical transformation using suitably functionalized small-fluorescent molecules or bioactive small molecules would be one example (among others) to demonstrate the scope and also addressing the utility of the His-Tag for purification purposes.

4. Furthermore, it is hard to establish from this approach a general research tool or biopharmaceutical, as it seems that the reactions for the given proteins are not optimized (see the comment in their rebuttal letter to question 2A of referee 1) and we still wonder what the final optimized conditions for proteins acylation are. The authors just stated their findings: when treating a "solution of GH6-EGFP... with 40 equiv. of 18 for 4 days, followed by the addition of two aliquots of 10 equiv. of 18 in the next two days, 88 % conversion was observed by ESI-MS (71 % conversion to mono-functionalized products)" and "when the protein was treated with 20 equiv. of 18 for 1 day, 51% was converted, of which 45% was mono-functionalized". In summary, we have the choice between good conversion (71%) but high amount of side products (17%), by using large excess of acylating reagent (60 equiv in total) and long reaction time (6 days in total) and modest conversion (45%), 5% side products and 1 day reaction time. Therefore, it is highly recommending to the authors to develop an optimized (high yield, less side products and shorter reaction time) for a given protein system, screening temperature, equivalents, buffers, sequences, etc. Along those lines, I am skeptical that a strategy, which delivers a mixture of modification products of a genetically engineered protein, would be very beneficial for the community, as many other methods exist, where genetically modified proteins deliver clean mono-functionalized products.

5. Finally, I thank the authors for their attempt to explain the design of their mechanistic studies and what a positive control is and I think that wording the changed wording of the sentence already helped to clarify their rationale. Nevertheless, in my opinion it would be recommendable to test the hypothetical formation of an N-Acyl derivative with the different activated 2-azidoacetic acids, in particular to obtain the NMR signal and help in the understanding of the reactivity of this N-acyl derivative, which could be crucial for the optimization of the reaction conditions (less side products, less equivalent of acylating reagent used and shorter reaction time etc...). As a minor comment, if I understand their experiments correctly the sentence "The hypothetical N-acyl derivatives of imidazole and Ac-GHHHHHH-NH₂ were not observed..." should be changed to "The hypothetical N-acyl derivatives of imidazole or Ac-GHHHHHH-NH₂ were not observed..."

We appreciate that the three reviewers have taken time to assess our revised manuscript *His Tag Acylation: Selective N-terminal acylation of peptides and proteins with an optimized His sequence by 4-methoxyphenyl esters*

We are very pleased with the comments by reviewers 1 and 2. The requests by reviewer #3 were quite extensive. We have worked hard to obtain new experimental results. Here is our detailed reply to reviewer #3.

2 The reviewer comments that we have only used few 4-methoxyphenyl esters for the chemical modification. In the paper we report that we can incorporate an azide moiety that then can be used to incorporate other entities such as PEG.

Reply

We have now synthesized and fully characterized a new 4-methoxyphenyl ester of a biotin reagent. We used this new reagent to introduce biotin onto SUMO and a new protein, BIR-2. Furthermore, we also developed an efficient protocol for the introduction of a fluorophore, which will be useful when working with costly fluorophores.

#3 Here the reviewer asks for

“application of a protein is given in which the functionality of the modified protein is critically addressed or the benefit of the particular modification strategy becomes visible”.

Reply

Our manuscript reports a new methodology, which we believe can have many applications. We have added a fourth protein to our study, a fragment of XIAP containing the BIR2 domain. The BIR2 domain was expressed with the required *N*-terminal Gly-His6 tag. Then, we biotinylated it with the new biotin 4-methoxyphenyl ester with. In addition, we coupled a fluorophore to BIR2 which was

first modified with 4-methoxyphenyl azido acetate. This demonstrates the generality and applicability of our His Tag acylation method.

We have in two cases shown that the N-terminal modification by our His Tag acylation method is compatible with the functionality of the proteins.

The first step in SUMOylation is C-terminal processing by SUMO protease to reveal a C-terminal Gly, which is essential for subsequent activation to form the thioester. The processing requires recognition and cleavage by SUMO protease. We have demonstrated that biotinylated SUMO can still be processed by SUMO protease. This result strongly suggests that also other enzymes involved in the SUMOylation process will still recognize the modified SUMO and, hence, that our His Tag acylation method can be used in assays.

Secondly, the XIAP fragment that we used is known to inhibit caspase-7. We have demonstrated that after N-terminally labelling with azidoacetyl it retains the ability to inhibit caspase-7.

Reviewer 3 also writes:

“The acylation of therapeutically or biologically relevant proteins followed by their chemical transformation using suitably functionalized small-fluorescent molecules or bioactive small molecules would be one example (among others) to demonstrate the scope and also addressing the utility of the His-Tag for purification purposes.”

Reply

In fact, we already use our new His Tag for purification. We mention this in the paper (Page 3, 15, and most explicitly on page 21). We acknowledge that our statements on page 3 and 15 could be clearer. The reviewer might think that we merely *suggest* that our new His tag can be used for purification purposes. We have now clarified in the text that we do use the dual functionality of the tag, hence making visible the benefit of our modification strategy, in accordance to what the reviewer asks for. This was already the case for EGFP and we also used His tag purification for the new protein, BIR-2.

4 Here the reviewer requests further optimization of our His Tag Acylation method.

The reviewer writes that

“...I am skeptical that a strategy, which delivers a mixture of modification products of a genetically engineered protein, would be very beneficial for the community, as many other methods exist, where genetically modified proteins deliver clean mono-functionalized products.”

Reply

We discuss the most important current methods for N-terminal modification of proteins. We are wondering what the reviewer has in mind regarding “many other methods exist, where genetically modified proteins deliver clean mono-functionalized products.”

We can only think of the following methods, all of which have limitations:

Incorporation of Cys for subsequent reaction at the free thiol: This is a very established method, however, its limitations include instability of some proteins with an additional Cys, post-translational modification of the Cys thiol, reversibility of the maleimide coupling (lack of stability), and more.

Ser oxidation and related methods: These are useful methods, however, Ser oxidation requires a rather strong oxidant, which can oxidize Met residues and can lead to loss of protein function.

N-terminal Cys for native chemical ligation: This is a versatile reaction, however, it requires a thioester for coupling, high concentrations, and often the need to eliminate or alkylate the thiol functionality after coupling, and more.

Non-natural amino acid incorporation: This is very elegant but it often gives low protein yields and the requirement to add the non-natural amino acid to the medium makes it an expensive technique, which is unattractive when large amounts of protein are required.

Enzymatic approaches, for example, with sortase: While these are useful methods they do require an enzyme, which often has to be added in relatively large amounts, requires an additional purification step afterwards, and is either expensive (in case the enzyme is purchased) or more labor-intensive (in case the enzyme is produced and purified in-house).

We believe that our method, as stated in the paper, has clear advantages as it, for example, does not require a strong oxidant (as is used for oxidation of an N-terminal Ser). We find that reviewer 3 disregards the limitations of current methods for chemical modification of proteins.

We have expanded our study on the optimization of the acylation of SUMO with the ester **18**. This is shown in a new table in the supporting information. This new table shows how the acylation of

protein, in this case SUMO, can be optimized. Importantly, four separate conditions gave N-terminal mono-acylation in > 70 % yield. The fact that several conditions gave high yields demonstrated the generality of the new His Tag acylation method. The best conditions gave a yield of 80%, which was higher than the already very good yield of 74% that we reported in the first version of this manuscript. This clearly demonstrates how the yield can be optimized ('tweaked').

We believe it is fair to compare our new method with previously reported methods. For example, in the state-of-the-art paper by MacDonald et al. they use 2-PCA for N-terminal modification (*Nature Chem Biol* 2015, 10.1038/NCHEMBIO.1792). Their degrees of conversion varied from 43 to 96% (9 model proteins), with most values (6 model proteins) between 63 and 82%, and 67% as average. It appears (Supplementary Fig 9) that at least some of the imidazolidinone products were unstable and partially hydrolyzed back (incubation at 37 °C, 12 h, 20-30% decrease in modified protein). We believe that our method compares favorably with this state-of-the-art method published in *Nature Chemical Biology*.

5 Reviewer 3 writes:

"... in my opinion it would be recommendable to test the hypothetical formation of an N-Acyl derivative with the different activated 2-azidoacetic acids, in particular to obtain the NMR signal and help in the understanding of the reactivity of this N-acyl derivative, which could be crucial for the optimization of the reaction conditions."

Reply

Here the reviewer suggests that we perform additional mechanistic studies with the phenyl and 4-nitrophenyl esters that we are *not* using but which were included in the development phase. We respectfully disagree with reviewer 3. We have included the explicit formation of the acyl imidazole as a positive control. We fail to see how the experiments suggested by the reviewer "could be crucial for the optimization of the reaction conditions". However, we performed the requested NMR experiments and included them in Figure SI 18.

REVIEWERS' COMMENTS:

Reviewer #3 (Remarks to the Author):

The authors have performed additional experiments, in particular a fourth protein, namely a fragment of X-linked inhibitor of apoptosis protein (XIAP) that contains the BIR2 domain, which was modified with 4-methoxyphenyl azido acetate. Furthermore, the later conjugate was coupled to a fluorophore through strain promoted Click chemistry. In addition, a new biotin 4-methoxyphenyl ester was prepared and used for the functionalization of the aforementioned protein fragment (XIAP) expanding further the scope of their methodology. Finally, the authors carried out an optimization study on the acylation of the GH6-SUMO protease affording a more comprehensive study on the behavior of their system. In principle, I can now support publication of this paper, especially, since the authors also demonstrated different strategies for optimizing the acylation protocol to a given protein of interest.

Nevertheless, I would like to add, that I don't think that comparing their study with the paper by MacDonald et al can be fully justified. In this study the Francis group has developed a new concept based on their own finding and has reported up to 5 different 2-PCA reagents (five steps synthesis for each) for the site-specific modification of peptides and proteins. Reagents functionalized with Fluorescein, MRI-contrasting chelators, biotin, PEG linker and the biologically relevant folic acid have been prepared. In addition, the site-specific biotinylation of 10 proteins with yields spanning between 43% and >95% and excellent selectivity (one example with double modification). On top of that, the authors have demonstrated the wide applicability of their methodology in challenging conditions namely the modification of proteins in the presence of free cysteines and at relevant pH values. In my humble opinion, the cited study is on another scientific level (also still considering that the current concept is based on established work). Along those lines, I am for sure not "disregarding the limitations of current methods for chemical modification of proteins", I just simply think that the current protocol cannot hold up protein modification strategies, which deliver homogeneously modified products.

Minor comments, that still need to be addressed:

1) "To further evaluate the generality of the selective acylation of the N-terminal segment, we reacted GH6-tagged MBP with 4-methoxyphenyl ester 18. Rewardingly, under the conditions tested mono-functionalized products were the major species. Conversion to mono-labeled protein was determined by ESI-MS as 59 % for GH6-MBP (Figure 4E)." The authors should specify the conditions used?

2) "The multiple additions of ester 18 took its half-life of in aq. solution into consideration. Conversion to monolabeled GH6-SUMO was determined by ESI-MS to be up to 80 %." The authors should write again this sentence and specify the reaction conditions that afford the 80% conversion?

3) The optimization table S1 should indicate the reactions conditions (buffer, concentrations, amount of proteins, technique used for determination of the yields etc...)

Point-by-point response to reviewer #3

We are grateful that reviewer #3 took the time to read the new and expanded manuscript. We are pleased that that the reviewer now writes:

“In principle, I can now support publication of this paper, especially, since the authors also demonstrated different strategies for optimizing the acylation protocol to a given protein of interest.”

Response

1) “To further evaluate the generality of the selective acylation of the N-terminal segment, we reacted GH6-tagged MBP with 4-methoxyphenyl ester 18. Rewardingly, under the conditions tested mono-functionalized products were the major species. Conversion to mono-labeled protein was determined by ESI-MS as 59 % for GH6-MBP (Figure 4E).” The authors should specify the conditions used?

We have done so

2) “The multiple additions of ester 18 took its half-life of in aq. solution into consideration. Conversion to monolabeled GH6-SUMO was determined by ESI-MS to be up to 80 %.” The authors should write again this sentence and specify the reaction conditions that afford the 80% conversion?

We have done so

3) The optimization table S1 should indicate the reactions conditions (buffer, concentrations, amount of proteins, technique used for determination of the yields etc...)

We have done so